# Multi-Modal Foundation Models for Computational Pathology: A Survey

**Dong Li**[*]                                                              *dong_li1@baylor.edu*
*Department of Computer Science, Baylor University*

**Guihong Wan**[*]                                                          *gwan@mgh.harvard.edu*
*Department of Dermatology, Massachusetts General Hospital, Harvard Medical School*

**Xintao Wu**                                                               *xintaowu@uark.edu*
*Electrical Engineering and Computer Science Department, University of Arkansas*

**Xinyu Wu**                                                                *xinyu_wu1@baylor.edu*
*Department of Computer Science, Baylor University*

**Xiaohui Chen**                                                            *xiaohui_chen1@baylor.edu*
*Department of Computer Science, Baylor University*

**Yi He**                                                                   *yihe@wm.edu*
*Department of Data Science, The College of William and Mary*

**Zhong Chen**                                                              *zhong.chen@siu.edu*
*School of Computing, Southern Illinois University*

**Peter K. Sorger**                                                         *peter_sorger@hms.harvard.edu*
*Department of Systems Biology, Harvard Medical School*

**Chen Zhao**[†]                                                            *chen_zhao@baylor.edu*
*Department of Computer Science, Baylor University*

**Reviewed on OpenReview:** *https://openreview.net/forum?id=NZ7GSH92cY*

## Abstract

Foundation models have emerged as a powerful paradigm in computational pathology (CPath), enabling scalable and generalizable analysis of histopathological images. While early developments centered on uni-modal models trained solely on visual data, recent advances have highlighted the promise of multi-modal foundation models that integrate heterogeneous data sources such as textual reports, structured domain knowledge, and molecular profiles. In this survey, we provide a comprehensive and up-to-date review of multi-modal foundation models in CPath, with a particular focus on models built upon hematoxylin and eosin (H&E) stained whole slide images (WSIs) and tile-level representations. We categorize 34 state-of-the-art multi-modal foundation models into three major paradigms: vision-language, vision-knowledge graph, and vision-gene expression. We further divide vision-language models into non-LLM-based and LLM-based approaches. Additionally, we analyze 30 available multi-modal datasets tailored for pathology, grouped into image-text pairs, instruction datasets, and image-other modality pairs. Our survey also presents a taxonomy of downstream tasks, highlights training and evaluation strategies, and identifies key challenges and future directions. We aim for this survey to serve as a valuable resource for researchers and practitioners working at the intersection of pathology and AI.

---

[*]Equal Contribution
[†]Corresponding Author

# 1   Introduction

The advent of foundation models has significantly transformed computational pathology (CPath) by enabling scalable and generalizable deep learning solutions for analyzing histopathological images. These models are designed to extract meaningful patterns from vast collections of pathological data, enhancing diagnostic accuracy, prognostic assessments, and biomarker discovery (Ochi et al., 2025). Among various imaging modalities, hematoxylin and eosin (H&E) stained images remain the most widely used in CPath due to their accessibility and effectiveness in capturing morphological details of tissues (Chanda et al., 2024; Guan et al., 2025). Whole Slide Images (WSIs), obtained from high-resolution scanning of tissue samples, offer comprehensive histopathological insights but are computationally demanding due to their large size. To manage this, WSIs are typically divided into smaller tile images, which serve as the fundamental units for training deep learning models (Wang et al., 2024; Ding et al., 2024; Chen et al., 2024d). Existing foundation models for CPath (FM4CPath) can be broadly classified into uni-modal and multi-modal paradigms (Li et al., 2025a), with the former primarily focusing on visual representation learning and the latter integrating additional modalities such as text, knowledge graphs, and gene expression profiles for enhanced interpretability and performance.

Early research in CPath predominantly leveraged uni-modal foundation models (Wang et al., 2022b; Chen et al., 2024c; Vorontsov et al., 2023), where deep learning models were trained solely on histopathological images. These uni-modal models have led to significant advancements in classification, segmentation, and prognostic prediction tasks by learning rich visual features from pathology slides. However, despite their success, these models are inherently limited by their exclusive reliance on image data, which often lacks crucial contextual information present in pathology reports, structured knowledge, or molecular profiles. To overcome these limitations, recent efforts have shifted toward multi-modal foundation models (Lu et al., 2024a; Wang et al., 2024; Lu et al., 2024b), which integrate heterogeneous data sources to provide more robust and interpretable insights.

Existing multi-modal foundation models for CPath (MMFM4CPath) can be categorized into three primary paradigms: vision-language, vision-knowledge graph, and vision-gene expression models. A roadmap of up-to-date MMFM4CPath is shown in Figure 1. Vision-language models (Huang et al., 2023; Ikezogwo et al., 2024; Sun et al., 2024e;b) utilize textual annotations, such as WSI reports and tile-level captions, to enrich visual representations, facilitating zero-shot learning and seamless cross-modal integration between images and text. Within this category, models can be further divided into non-LLM-based and LLM-based approaches, with the latter incorporating large language models (LLMs) for improved natural language understanding and generative capabilities. Vision-knowledge graph models (Zhou et al., 2024b;a) integrate structured domain knowledge by leveraging pathology-specific ontologies and knowledge graph to guide deep learning models. Vision-gene expression models (Xu et al., 2024; Vaidya et al., 2025) align visual features with molecular-level insights from RNA sequencing and other omics data, facilitating genotype-phenotype associations for precision medicine.

While existing surveys have explored FM4CPath (Ochi et al., 2025; Chanda et al., 2024; Guan et al., 2025; Bilal et al., 2025; Li et al., 2025a), they often lack a comprehensive analysis tailored to multi-modal approaches. As shown in Table 1, our survey differentiates itself by systematically categorizing 34 of the most up-to-date MMFM4CPath and analyzing 30 available multi-modal datasets for pathology, with an emphasis on modalities beyond vision-language integration. Additionally, we provide an in-depth discussion on evaluation methodologies, training strategies, and emerging challenges in this field. For transparency, we detail our literature search strategy, including databases, search terms, time range, and inclusion/exclusion criteria, in Appendix A. The key contributions of this survey include:

- **Comprehensive and Up-to-Date Survey.** This survey systematically reviews 34 multi-modal foundation models in computational pathology across vision-language, vision-knowledge graph, and vision-gene expression paradigms. It offers detailed comparisons of their architectures, pretraining strategies, and adaptation techniques, providing a broader and more current coverage than prior surveys.

- **In-Depth Analysis of Pathology-Specific Multi-Modal Datasets.** This survey curates and categorizes 30 available datasets into three types: image-text pairs, multi-modal instructions, and image-other

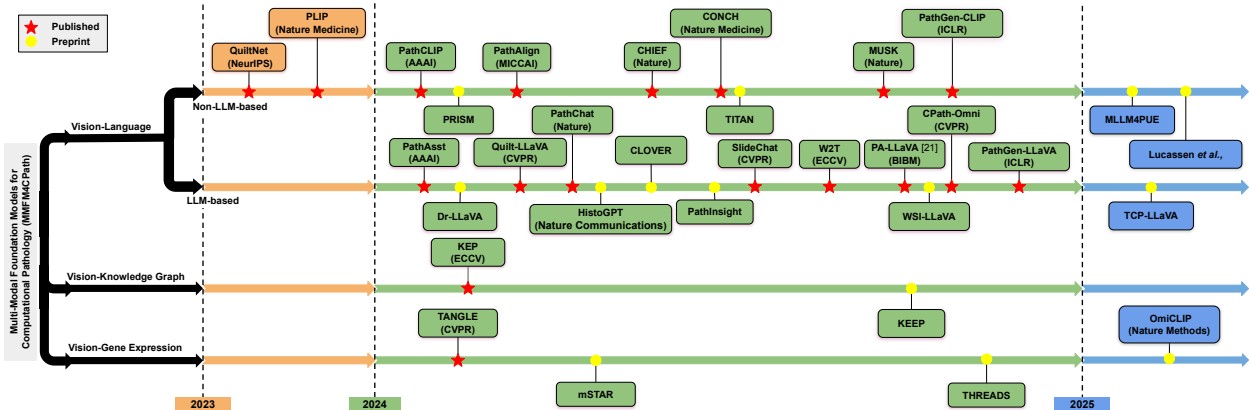

Figure 1: A roadmap of multi-modal foundation models for computational pathology (MMFM4CPath).

Table 1: Comparison between our survey and related surveys.

| Survey | # MMFM4CPath | | | | | # Datasets for MMFM4CPath | | | | Tasks Taxonomy |
|---|---|---|---|---|---|---|---|---|---|---|
| | Vision-Language | | Vision-Knowledge Graph | Vision-Gene Expression | Total | Image-Text Pair | Instruction | Image-Other Modality | Total | |
| | Non-LLM | LLM | | | | | | | | |
| Ochi *et, al.* (Ochi et al., 2025) | 4 | ✗ | ✗ | 1 | 5 | 4 | ✗ | 1 | 5 | ✓ |
| Chanda *et, al.* (Chanda et al., 2024) | 7 | 4 | 1 | ✗ | 12 | 8 | 6 | ✗ | 14 | ✗ |
| Guan *et, al.* (Guan et al., 2025) | 3 | 11 | ✗ | 1 | 14 | 8 | 6 | ✗ | 14 | ✗ |
| Bilal *et, al.* (Bilal et al., 2025) | 8 | 4 | ✗ | 2 | 14 | ✗ | ✗ | ✗ | ✗ | ✓ |
| Li *et, al.* (Li et al., 2025a) | 8 | ✗ | 2 | ✗ | 10 | 12 | ✗ | 2 | 14 | ✓ |
| This Survey | 13 | 15 | 2 | 4 | 34 | 12 | 13 | 5 | 30 | ✓ |

modality pairs. We emphasize how these datasets enable various training strategies and highlight their roles in aligning modalities and supporting instruction tuning.

- **Thorough Overview of Multi-Modal Evaluation Tasks.** A taxonomy of evaluation tasks is provided, covering six major categories including classification, retrieval, generation, segmentation, prediction, and visual question answering. We detail how different MMFM4CPath are evaluated under various settings.

- **Future Research Opportunities.** We outline three promising directions, such as integrating H&E images with spatial omics data for deeper biological insight, leveraging H&E to predict MxIF markers for cost-effective virtual staining, and establishing standardized benchmarks to ensure consistent evaluation across tasks and datasets. These directions aim to enhance the clinical relevance, scalability, and comparability of future models.

# 2 Background

## 2.1 Computational Pathology

Computational Pathology (CPath) is an interdisciplinary field that applies computational techniques, including machine learning and computer vision, to analyze and interpret pathological data. By leveraging digital pathology, CPath enhances diagnostic accuracy, facilitates large-scale biomarker discovery, and supports personalized medicine. Among the various imaging modalities in pathology, Hematoxylin and Eosin (H&E) stained images serve as the most commonly used vehicle for studying CPath. These images capture essential morphological characteristics of tissues, making them fundamental for histopathological analysis. Within the realm of digital pathology, Whole Slide Images (WSIs) and tile images are two primary forms of data representation. WSIs, generated from high-resolution scanning of entire tissue slides, provide comprehensive visual information at gigapixel scale, allowing pathologists to examine cellular structures in detail. However, due to their enormous size and high computational demands, WSIs pose significant challenges in terms of storage, processing, and analysis. To mitigate these challenges, WSIs are often divided into smaller,

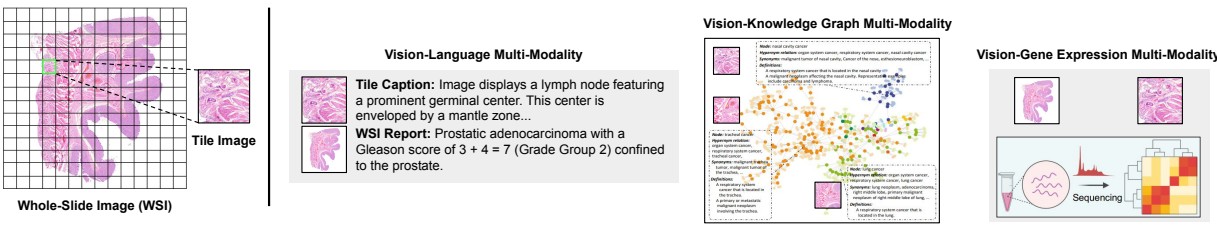

Figure 2: (Left) Illustration of whole-slide image and its corresponding tile images from H&E-stained tissue. (Right) The three primary types of multi-modal approaches in computational pathology.

more manageable tile images, which serve as the primary unit of analysis in many computational pathology studies.

While visual analysis remains central to CPath, researchers increasingly rely on multi-modal data to enhance interpretability and improve model performance. One major auxiliary modality is language, which includes both tile-level captions that describe specific regions of tissue and WSI-level pathology reports that provide global contextual information about a slide. Integrating text data with images enables vision-language models to learn richer feature representations and facilitate interpretability. Another important modality is structured domain knowledge, often represented in knowledge graphs, which encode relationships between diseases, biomarkers, and tissue structures, guiding AI models toward more biologically plausible interpretations. Additionally, molecular data, such as gene expression profiles, offer complementary insights by linking histopathological features to underlying genetic mechanisms. By aligning visual data with gene expression information, vision-gene expression models enable the discovery of novel genotype-phenotype associations. Figure 2 illustrates examples of WSIs and tile images alongside the three major multi-modal paradigms in CPath. The synergy of these multi-modal approaches, including vision-language, vision-knowledge graph, and vision-gene expression, has proven crucial in advancing the field of CPath, enabling more robust, generalizable, and interpretable AI-driven pathology models.

## 2.2 Pre-training Objective for Multi-Modal FMs

Unlike uni-modal models, which are primarily pre-trained through self-supervised contrastive learning (SSCL), multi-modal FMs, due to their cross-modal nature, involve a more diverse set of self-supervised learning (SSL) objectives during their pre-training process. Furthermore, when fine-tuning LLMs to enable conversational abilities, supervised instruction tuning is usually required.

The primary pre-training objective for multi-modal FMs is SSCL. CLIP (Radford et al., 2021), as a pioneer in this field, ensures that the embeddings generated by the image encoder and text encoder are as similar as possible for paired image-text data by utilizing contrastive loss. CoCa (Yu et al., 2022) builds upon CLIP by adding a multi-modal encoder and an additional captioning loss to enable the mapping from the visual space to the language space. BLIP-2 (Li et al., 2023) trains a lightweight Querying Transformer (Q-Former) using a two-stage strategy. In the first stage, a frozen image encoder bootstraps vision-language representation learning, while in the second stage, visual features are mapped to the language model input space, leveraging a frozen LLM for text generation. Additionally, next word prediction (NWP) is a text-specific SSL task commonly used for fine-tuning LLMs. It aims to predict the most likely next token based on the given text sequence. Moreover, cross-modal alignment (CMA) multi-modal domain-specific task, which aims to build a unified semantic space where the embedding vectors from different modalities can reflect the same semantic content. In addition to contrastive learning, generative reconstruction and prediction are also commonly used SSL proxy tasks for CMA.

Instruction tuning (IT) is a method for fine-tuning LLMs to enable them to better understand and execute the instructions or task requirements provided by users. Unlike traditional pretraining objectives like NWP, the goal of IT is to enable the model to generate meaningful responses or actions based on specific instructions or questions. In Instruction Tuning, the model not only learns how to generate language but also learns how to adapt and generate different outputs according to various task requirements. This typically involves

supervised training using a large number of instructions, ensuring that the model can understand the intent of the tasks and effectively perform them. Such tasks can include text generation, question answering, and conversation.

# 3 Multi-Modal Foundation Models for CPath

The power of multi-modal data has been repeatedly validated not only in the general machine learning community (Wang, 2021; Wang et al., 2023; Wu et al., 2023) but also in the field of computational pathology (Ochi et al., 2025; Chanda et al., 2024; Guan et al., 2025). Given the high cost of pathology image data, leveraging other modalities, particularly textual data, as auxiliary information to learn more robust tile or WSI representations has become a dominant approach in developing foundation models for pathology (FM4CPath). Based on the modalities used, we categorize existing multi-modal FM4CPath (MMFM4CPath) into three major groups: vision-language, vision-knowledge graph, and vision-gene expression models. Additionally, we comprehensively summarize their network architectures and pre-training details across different stages, as shown in Table 2. The rapid advancements in LLMs have enabled MMFM4CPath to possess enhanced generation and conversational capabilities. We further categorize vision-language models into non-LLM-based and LLM-based approaches. The goal of these methods is typically to learn robust representations of tiles or WSIs for a wide range of downstream tasks.

## 3.1 Non-LLM-Based Vision-Language FM4CPath

Vision-language FM4CPath enhance the models' understanding of pathological images by aligning paired image-text data under vision-language SSL frameworks, such as CLIP (Radford et al., 2021) and CoCa (Yu et al., 2022), enabling them to learn robust visual representations while also supporting zero-shot and cross-modal tasks. These methods typically use an off-the-shelf or trained vision encoder before performing joint visual-language pre-training, which has been shown to improve the performance (Zimmermann et al., 2024). They also leverage existing LLMs or Visual LLM (V-LLMs, *a.k.a.* MLLMs), typically in two ways: (i) fine-tuning them to serve as text encoders, or (ii) leaving them untuned, solely utilizing their capabilities for generation and conversation.

**CLIP-based Vision-Language FM4CPath.** The success of CLIP on natural images has inspired some works to apply it in the CPath domain. PLIP (Huang et al., 2023), PathCLIP (Sun et al., 2024e) and QuiltNet (Ikezogwo et al., 2024) all fine-tune a CLIP model pre-trained on natural images using datasets composed of paired tiles and their captions. CHIEF (Wang et al., 2024) uses an image encoder pretrained for CPath domain (Wang et al., 2022b) to encode the tile sequence extracted from WSIs to obtain WSI-level features and CLIP's text encoder to encode anatomical site information (WSI-level label). A weakly supervised aggregation network then combines both modalities to generate rich multi-modal WSI representations. Unlike previous methods that rely on out-of-shelf vision encoders, PathGen-CLIP (Dai et al., 2024) leverages the generative capabilities of V-LLMs to obtain high-quality tile-caption pairs and uses them to train an OpenAI CLIP (Radford et al., 2021) framework from scratch, followed by fine-tuning on tile-caption pairs from public datasets. PathAlign-R (Ahmed et al., 2024) is also trained from scratch on pathology data using the CLIP framework, but it focuses on the WSI-level. MLLM4PUE (Zhou et al., 2025) leverages V-LLMs as the backbone to generate universal multi-modal embeddings for CPath, integrating images and text within a single framework to better understand their complex relationships.

**CoCa-based Vision-Language FM4CPath.** CoCa's multi-modal decoder serves as a crucial bridge between visual and linguistic information. By transforming encoded image features into text-aware representations, it significantly boosts the cross-modal integration of MMFM4CPath, thereby enhancing its performance in advanced pathology applications. CONCH (Lu et al., 2024a), PRISM (Shaikovski et al., 2024), and Lucassen *et al.* (Lucassen et al., 2025) all pre-train an image encoder on pathology datasets, and then further conduct joint vision-language pre-training within the CoCa framework. The difference is that PRISM and Lucassen *et al.* extend the image encoder to the WSI-level using a Perceiver network (Jaegle et al., 2021) and employ WSIs along with their corresponding clinical reports for training. MUSK (Xiang et al., 2025) first independently trains image and text encoders on unpaired pathology images and text tokens via masked data modeling within the BEiT-3 framework (Wang et al., 2022a). Using Masked Image

Table 2: Overview of architecture and pre-training details of MMFM4CPath (Due to space constraints, the references for the mentioned LLMs, V-LLMs, and off-the-shelf architectures are provided in the footnote of this table.)

| Group | Model (Availability) | Year | Vision (V)‡ | Language (L) / Knowledge Graph (KG) / Gene Expression (GE) | Multi-Modal | Objective¶ | V | O | M | Data Short Description | Input Image Type |
|---|---|---|---|---|---|---|---|---|---|---|---|
| Vision-Language / Non-LLM-Based | QuiltNet (Ikezogwo et al., 2024) ✓ | 2023 | T: ViT-B/32 | L: Transformer Layers | - | SSL (CLIP) | D | D | - | 438K Tiles and 802K Captions | Tiles |
| | PLIP (Huang et al., 2023) ✓ | 2023 | T: ViT-B/32 | L: Transformer Layers | - | SSL (CLIP) | D | D | - | 208K Tile-Caption Pairs | Tiles |
| | PathCLIP (Sun et al., 2024e) ✗ | 2024 | T: ViT-B/32 | L: Transformer Layers | - | SSL (CLIP) | D | D | - | 207K Tile-Caption Pairs | Tiles |
| | PRISM (Shaikovski et al., 2024) ✗ | 2024 | T: ViT-H/14, W: Perceiver Net. | L: BioGPT (L1–12) | BioGPT (L13–24) with Cross-Attention Layers | SSL (CoCa) | F,S | F | F,S | 587K WSIs with 195K Specimens | WSIs |
| | PathAlign-R (Ahmed et al., 2024) ✗ | 2024 | T: ViT-S/16, W: Q-Former | L: Q-Former | - | SSL (MSN) / SSL (CLIP) | S,N / F,S | N / S | - / - | Tiles From 354,089 WSIs / 434k WSI-Report Pairs | WSIs |
| | PathAlign-G (Ahmed et al., 2024) ✗ | 2024 | T: ViT-S/16, W: Q-Former | L: Q-Former, L (LLM): PaLM-2 S | MLP | SSL (MSN) / SSL (BLIP-2) / SSL (CMA) | S,N / F,S / F,D | N,N / S,N / N,F | N / N / S | Tiles From 354,089 WSIs / 434k WSI-Report Pairs / - | WSIs |
| | CHIEF (Wang et al., 2024) ✓ | 2024 | T: Swin-T, W: Aggregator Net. | L: Transformer Layers | MLP | WSL (CLIP) | D,S | D | S | 60K WSIs with Labels | WSIs |
| | CONCH (Lu et al., 2024a) ✓ | 2024 | T: ViT-B/16 | L: Transformer Layers | Transformer Layers | SSL (iBOT) / SSL (NWP) / SSL (CoCa) | S / N / D | N / S / D | N / S / D | 16M Tiles Sampled From 21K WSIs / >950K Pathology Text Entries / 1.17M Tile-Caption Pairs | Tiles |
| | TITAN (Ding et al., 2024) ✓ | 2024 | T: ViT-L, W: ViT-S | L: Transformer Layers | Transformer Layers | SSL (iBOT) / SSL (CoCa) / SSL (CoCa) | F,S / F,D / F,D | N / D / D | N / D / D | 336K WSIs / 423K ROI-Caption Pairs / 183K WSI-Report Pairs | WSIs |
| | MUSK (Xiang et al., 2025) ✓ | 2025 | T: V-FFN | L: L-FFN \|←— Shared Attention Layers —→\| | Cross-Attention Decoder | SSL (BEiT3) / SSL (CoCa) | S / D | S / D | N / S | 1B Text Tokens and 50M Tiles / 1.01M Tile-Caption Pairs | Tiles |
| | PathGen-CLIP (Sun et al., 2024d) ✗ | 2025 | T: ViT-B/32 | L: Transformer Layers | - | SSL (CLIP) / SSL (CLIP) | S / D | S / D | - / - | 1.6M High-Quality Tile-Caption Pairs / 700K Tile-Caption Pairs | Tiles |
| | MLLM4PUE (Zhou et al., 2025) ✗ | 2025 | T: SigLIP | L: Qwen 1.5 | MLP | SSL (CLIP) | D | D | D | 594K Tile-Caption Pairs | Tiles |
| | Lucassen et al. (Lucassen et al., 2025) ✗ | 2025 | T: ViT-L/14, W: Perceiver Net. | L: BioGPT (L1–12) | BioGPT (L13–24) with Cross-Attention Layers | SSL (CoCa) | F,S | F | F,S | 42K WSIs and 19K Reports | WSIs |
| Vision-Language / LLM-Based | PathAsst (Sun et al., 2024e) ✗ | 2024 | T: ViT-B/32 | L (LLM): Vicuna-13B | MLP | SSL (CMA) / SL (IT) | F / F | F / I | S / D | Description Part of PATHINSTRUCT / 35K Samples From PATHINSTRUCT | Tiles |
| | Dr-LLaVA (Sun et al., 2024a) ✓ | 2024 | T: ViT-L/14 | L (LLM): Vicuna-V1.5 | MLP | SL (IT) & RL | D | I | D | Multi-turn Dialogues Based on 16K Tiles | Tiles |
| | Quilt-LLaVA (Seyfioglu et al., 2024) ✓ | 2024 | T: ViT-B/32 | L (LLM): GPT-4 | MLP | SSL (CMA) / SL (IT) | F / F | F / I | S / D | 723K Tile-Caption Pairs / 107K Pathology-Specific Instructions | Tiles |
| | PathChat (Lu et al., 2024b) ✓ | 2024 | T: ViT-L/16 | L (LLM): Llama 2-13B | MLP with Attention Pooling | SSL (CoCa) / SSL (CMA) / SL (IT) | D / F / F | N / F / I | S / D / D | 1.18M Tile-Caption Pairs / ~100K Tile-Caption Pairs / 457K Instructions with 999K VQA Turns | Tiles |
| | HistoGPT-S/M (Tran et al., 2025) ✓ | 2024 | T: Swin-T, W: Perceiver Net. | L (LLM): BioGPT-B / BioGPT-L | - | WSL (MIL) / SSL (NWP) | F,S / F,F | F/F / D/D | - / - | 15.1K WSIs with 6.7K Patient-Level Labels / 15.1K WSI-Reports Pairs | Tiles |
| | HistoGPT-L (Tran et al., 2025) ✓ | 2024 | T: ViT-L/16, W: GCN | L (LLM): BioGPT-L | - | SSL (NWP) | F,S | S | - | 15.1K WSI-Reports Pairs | Tiles |
| | CLOVER (Chen et al., 2024a) ✓ | 2024 | T: EVA-ViT-G/14 | L: Q-Former L (LLM): Vicuna 7B / FlanT5XL | Q-Former MLP | SSL (BLIP-2) / SL (IT) | F / F | S,N/N / N,I/I | S,N / N,S | 438K Tiles and 802K Captions / 45K VQA Instructions | Tiles |
| | PathInsight (Wu et al., 2024) ✓ | 2024 | \|←— V-LLM: LLaVA / Qwen-VL-7B / InternLM —→\| | | | SL (IT) | | I / I / I | | 45K Instances Covering 6 Pathology Tasks | Tiles |
| | SlideChat (Chen et al., 2024d) ✓ | 2024 | T: ViT-B, W: LongNet | L (LLM): Qwen2.5-7B | MLP | SSL (CMA) / SL (IT) | F,S / F,D | F / I | S / D | 4.2K WSI-Report Pairs / 176K Instruction-Following VQA Pairs | WSIs |
| | W2T (Chen et al., 2024b) ✓ | 2024 | T: ViT-S / Res-ResNet-50 / HIPT, W: Transformer Layers | L: PubMedBERT / BioClinicalBERT / An Embedding Mapping | Transformer Layers | SSL (NWP) | T: F / W: S | D / D / S | S | 804 WSIs with 7.14K VQA Pairs | WSIs |
| | PA-LLaVA (Dai et al., 2024) ✓ | 2024 | T: ViT-B/32 | L (LLM): LLama3 with LoRA | Transformer Layers | SSL (CLIP) / SSL (CMA) / SL (IT) | D / F / F | F / I / I | F / D / D | 827K Tile-Caption Pairs / 518K Tile-Caption Pairs / 35.5K VQA Pairs | Tiles |
| | WSI-LLaVA (Liang et al., 2024) ✗ | 2024 | T: ViT-G/14, W: LongNet, MLP | L: Bio_ClinicalBERT, L (LLM): Vicuna-7b-v1.5 | MLP | SSL (CLIP) / SSL (CMA) / SL (IT) | F,F,S / F,F,F / F,F,F | D,N / N,F / N,I | N / S / D | 9.85K WSI-Report Pairs / 9.85K WSI-Report Pairs / 175K VQA Pairs | WSIs |
| | CPath-Omni (Sun et al., 2024b) ✗ | 2024 | T: ViT-H/14, ViT-L, W: SlideParser | L: Qwen2.5-14B | MLP | SSL (CMA) / SL (IT) / SSL (CoCa) / SL (IT) | F,F,F / D,D,D / F,F,D / D,D,D | F / I / F / I | S / D / F / D | 700K Tile-Caption Pairs / 352K Tile Instructions / 5.85K WSI-Report Pairs / 53K Tile and 34K WSI Instructions | Tiles or WSIs |
| | PathGen-LLaVA (Sun et al., 2024d) ✗ | 2025 | T: ViT-B/32 | L: Transformer Layers, L (LLM): Vicuna | MLP | SSL (CLIP) / SSL (CMA) / SL (IC) | S / F / F | S,N / N,F / N,D | N / S / D | 700K Tile-Caption Pairs / 700K Tile-Caption Pairs / 30K Detailed Tile Descriptions | Tiles |
| | TCP-LLaVA (Lyu et al., 2025) ✗ | 2024 | T: ViT-B/16 | L: Qwen2.5-7B-Instruct | MLP | SL (IT) | F | F | I | 175K VQA Pairs | WSIs |
| Vision-KG | KEP (Zhou et al., 2024b) ✓ | 2024 | T: ViT-B/(16,32) | L: PubMedBERT ↩, KG: PubMedBERT | - | SSL (PKE) / SSL (CLIP) | N / D | N,S / D,F | - / - | A Pathology KG with 50.5K Attributes / 715K Tile-Caption Pairs | Tiles |
| | KEEP (Zhou et al., 2024a) ✓ | 2024 | T: ViT-L/16 | L, KG: PubMedBERT | - | SSL (PKE) / SSL (CLIP) | N / D | S / D | - / - | A Pathology KG with 139K Attributes / 143K Semantic Groups Through KG | Tiles |
| Vision-GE | TANGLE (Jaume et al., 2024) ✓ | 2024 | T: ViT-B (Rat) / Swin-T (Human) W: ABMIL | GE: A Three-Layer MLP | - | SSL (iBOT) / SSL (CLIP) | S/N,N / F/F,S | N / S | - / - | 15M Rat Tiles From 47K WSIs / 8.67K WSI- Gene Pairs | WSIs |
| | mSTAR (Xu et al., 2024) ✗ | 2024 | T: ViT-L/16, W: Two-Layer TransMIL | L: BioBERT-Basev1.2, GE: scBERT | - | SSL (CLIP) / SSL (SD) | F,S / D,F | D,D / N,N | - / - | 7.95K WSI-Report-Gene pairs / 7.95K WSIs | WSIs |
| | THREADS (Vaidya et al., 2025) ✗ | 2025 | T: ViT-L, W: ABMIL | GE: scGPT (RNA), A Four-Layer MLP (DNA) | - | SSL (CLIP) | F,S | D,S | - | 26.6K WSI-Gene (RNA) Pairs & 20.5K WSI-Gene (DNA) Pairs | WSIs |
| | OmiCLIP (Chen et al., 2025) ✓ | 2025 | T: ViT-B/16 | GE: MLP | MLP | SSL (CLIP) | F | S | D | 2.18M tile–transcriptomics caption pairs | Tiles |

† Network architecture types: **T**: Tile Encoder, **W**: WSI Encoder, **L**: Text Encoder, **LLM**: Large Language Model, **V-LLM**: Multi-modal LLM, **KG**: Knowledge Graph Encoder, **GE**: Gene Expression Encoder.

§ Multi-modal foundation models are typically pretrained in multiple stages, with **each row in this column representing a distinct pretraining phase**.

¶ Training objectives are categorized into Supervised Learning (**SL**), Weakly Supervised Learning (**WSL**), Self-Supervised Learning (**SSL**), and Reinforcement Learning (**RL**). SL includes Image Captioning (**IC**) and Instruction Tuning (**IT**), WSL includes Multiple Instance Learning (**MIL**), and SSL encompasses Contrastive Learning (**CL**), Masked Siamese Networks (**MSN**), Next Word Prediction (**NWP**), Cross-Modal Alignment (**CMA**), Pathology Knowledge Encoding (**PKE**), and Self-Distillation (**SD**). CL is further divided based on its contrastive objectives into **CLIP, CoCa, BLIP-2, iBOT** and **BEiT3**.

* Pre-training strategies for different architectures (**V**: Vision, **O**: Other Modalities, **M**: Multi-Modal): **F**: Frozen, **S**: From Scratch, **D**: Domain-Specific Tuning, **I**: Instruction Tuning, **N**: Not Used, -: Not Existed.

• References of mentioned LLMs and V-LLMs in Table 2: BioGPT (Luo et al., 2022), PaLM-2 S (Anil et al., 2023), Qwen 1.5 (Bai et al., 2023), Vicuna-13B (Chiang et al., 2023), Vicuna-V1.5 (Touvron et al., 2023a), GPT-4 (Achiam et al., 2023), Llama 2-13B (Touvron et al., 2023b), Vicuna 7B (Chiang et al., 2023), FlanT5XL (Chung et al., 2024), LLaVA (Liu et al., 2023), Qwen-VL-7B (Bai et al., 2023), InternLM (Zhang et al., 2023a), Qwen2.5-7B (Yang et al., 2024), LLama3 (Grattafiori et al., 2024), LoRA (Hu et al., 2022), Vicuna-7b-v1.5 (Zheng et al., 2023), Qwen2.5-14B (Hui et al., 2024), Vicuna (Chiang et al., 2023)

• References of mentioned off-the-shelf architectures in Table 2: Perceiver Net. (Jaegle et al., 2021), GCN (Gindra et al., 2024), EVA-ViT-G/14 (Fang et al., 2023), Q-Former (Li et al., 2023), Swin-T (Liu et al., 2021), V-FFN (Shazeer et al., 2017), L-FFN (Shazeer et al., 2017), SigLIP (Zhai et al., 2023), LongNet (Ding et al., 2023), PubMedBERT (Gu et al., 2021), BioClinicalBERT (Gu et al., 2021), HIPT (Chen et al., 2022), ABMIL (Ilse et al., 2018), TransMIL (Shao et al., 2021), BioBERT-Basev1.2 (Lee et al., 2020), scGPT (Cui et al., 2024), scBERT (Yang et al., 2022)

Modeling (MIM) (He et al., 2022), it leverages ViT's patch structure to predict missing patches and learn robust representations, and then aligns the two modalities within the CoCa framework. TITAN (Ding et al., 2024) proposes a novel foundational framework for whole-slide imaging analysis through three progressive

training stages: Initially, the WSI encoder is optimized via the iBOT framework (Zhou et al., 2021) enhanced with positional encoding; this is followed by a dual-scaled refinement under the CoCa framework leveraging tile-level features and WSI-level contexts, where pathology-specialized V-LLMs generate diagnostic captions and structured reports.

**Other Vision-Language FM4CPath.** Unlike previous methods that use CLIP or CoCa framework, PathAlign-G (Ahmed et al., 2024) first pre-trains a ViT-S using Masked Siamese Networks (MSN) (Assran et al., 2022), and then fine-tunes the model using the BLIP-2 framework. This enables PathAlign-G to utilize a shared pathology image-text embedding space, enhancing its cross-modal capabilities and making it more suitable for generative tasks.

### 3.2 LLM-Based Vision-Language FM4CPath

The powerful reasoning ability (Li et al., 2025b; Zhao et al., 2025) and the fusion of vision and language modalities (Radford et al., 2021; Yu et al., 2022) provides an extra perspective for MMFM4CPath, where pathological visual representations aligned with language signals in latent space can assist LLMs in understanding pathology knowledge, thereby contributing to the construction of generative foundation AI assistants for pathologists (Lu et al., 2024b). These methods acquire pathology-specialized V-LLMs by pairing a pre-trained image encoder with an LLM via a simple multi-modal module for cross-modal feature alignment, then fine-tuning the LLM. Beyond contrastive learning, they employ diverse pre-training objectives, from supervised to self-supervised learning. We classify them into instruction-based and non-instruction-based methods based on LLM fine-tuning approaches.

**Instruction-Based V-LLMs for CPath.** Most V-LLMs for CPath undergo instruction tuning on carefully curated datasets, refining general-purpose LLMs for the pathology domain while enhancing their cross-modal understanding. PathAsst (Sun et al., 2024e) builds a pathological V-LLM using PathCLIP as the visual backbone. It aligns the image encoder with the LLM via a trained layer on QA-based instructions, then fine-tunes the LLM with limited instructions. Following the same pre-training process, Quilt-LLaVA (Seyfioglu et al., 2024) and PA-LLaVA (Dai et al., 2024) are fine-tuned on their publicly available instruction-tuning datasets, while PathChat (Lu et al., 2024b) undergoes instruction tuning on its carefully designed and diverse instructions. SlideChat (Chen et al., 2024d) and WSI-LLaVA (Liang et al., 2024) go beyond the tile-level and create V-LLMs capable of handling gigapixel WSIs, and are fine-tuned on corresponding WSI-level instruction datasets. TCP-LLaVA (Lyu et al., 2025) is the first pathology WSI VQA model based on token compression. By introducing a modality compression module with trainable compression tokens, it compresses tens of thousands of patch tokens into a few hundred before feeding them into the LLM, significantly reducing computational cost while maintaining accuracy.

Instead of relying on separate image encoders or vision-language architectures, PathInsight (Wu et al., 2024) directly fine-tunes existing V-LLMs using instructions covering six pathology tasks. In addition to instruction tuning, Dr-LLaVA (Sun et al., 2024a) employs reinforcement learning (RL) with an automated reward function that assesses the clinical validity of responses during multi-turn interactions. CLOVER (Chen et al., 2024a) aims to develop a cost-effective V-LLM for conversational pathology. It employs BLIP-2 with a lightweight Q-former (Li et al., 2023), keeping both the visual encoder and LLM frozen to avoid full LLM tuning. CLOVER combines generation-based instructions from GPT-3.5 (Achiam et al., 2023) with template-based instructions, forming hybrid instructions that improve understanding. As one of the most powerful models currently, CPath-Omni (Sun et al., 2024b) aims to build a unified model that can process tile-level and WSI-level inputs separately through a proprietary framework, and integrate LLMs to enable generation and conversational capabilities. It undergoes four stages of training on three proposed datasets of different types: tile-caption pairs, tile-level instructions, and WSI-level instructions.

**Non-Instruction-Based V-LLMs for CPath.** PathGen-LLaVA (Sun et al., 2024d) is trained from scratch on the CLIP architecture using tile-caption pairs, then a fully connected (FC) layer is trained to ensure the features extracted by the image encoder are understandable by the LLM. Finally, it employs a supervised image captioning task rather than instruction tuning, as PathGen-LLaVA is specifically designed for generating pathology image descriptions. W2T (Chen et al., 2024b) utilizes four frozen visual extractors (including those trained on natural images and pathology images) and three text extractors in various

combinations. It is trained on its proposed WSI-VQA instruction dataset using next word prediction (NWP) to interpret WSIs through generative visual question answering. HistoGPT (Tran et al., 2025) is designed with three model sizes: small, medium, and large. Among them, HistoGPT-S and HistoGPT-M first train a Perceiver Network (Jaegle et al., 2021) as a WSI encoder using multiple instance learning (MIL), followed by fine-tuning the LLM with NWP. HistoGPT-L, on the other hand, employs a graph convolutional network (GNN) (Kipf & Welling, 2016) to encode WSI-level positional information, eliminating the need for a pre-trained WSI encoder. HistoGPT is capable of simultaneously generating reports from multiple pathology images and provides prompts that allow for expert knowledge guidance.

### 3.3 Enhancing FM4CPath with Other Modalities

Due to the high costs, pathology-specific datasets are typically small and sourced from diverse origins, such as websites or videos (Huang et al., 2023; Ikezogwo et al., 2024). This often results in noisy data with limited quality, making it unstructured and lacking domain knowledge. Meanwhile, massive multi-modal data aligned with clinical practices, along with domain-specific knowledge, such as gene expression profiles, remain underutilized for pretraining. Based on these, some studies have explored incorporating modalities beyond vision and language to enhance the training signal.

**Vision-Knowledge Graph FM4CPath.** To integrate structured domain-specific knowledge, KEP (Zhou et al., 2024b) constructs a pathology knowledge graph and encodes it using a knowledge encoder, which then guides vision-language pretraining. They design a pathology knowledge encoding (PKE) method to align semantic groups in the latent space for training the knowledge encoder. Similarly, KEEP (Zhou et al., 2024a) builds a disease knowledge graph for encoding and employs knowledge-guided dataset structuring to generate tile-caption pairs for pretraining within the CLIP framework, incorporating strategies such as positive mining, hardest negative sampling, and false negative elimination.

**Vision-Gene Expression FM4CPath.** Serving as WSI-level information, gene expression profiles provide insights into quantitative molecular dynamics, complementing the qualitative morphological perspective of a WSI and capturing biologically and clinically significant details. TANGLE (Jaume et al., 2024) is a transcriptomics-guided WSI representation learning framework that aligns image signals with RNA sequences encoded by multi-layer perceptron (MLP) in the latent space using contrastive loss, similar to CLIP. It extends beyond human tissues, incorporating a specialized architecture and dataset for rat tissue pretraining. THREADS (Vaidya et al., 2025), like TANGLE, utilizes molecular profiles from next-generation sequencing for WSI representation learning but uniquely integrates WSI-RNA and WSI-DNA sequence pairs. mSTAR (Xu et al., 2024) integrates three modalities within an extended CLIP framework, training on WSI-report-gene expression pairs via inter-modality and inter-cancer contrastive learning. It then employs self-distillation to transfer multi-modal knowledge to the patch extractor. Recently, OmiCLIP (Chen et al., 2025) is a visual–omics foundation model that builds on CoCa-style cross-modal contrastive learning to align H&E slides with spatial transcriptomics data, enabling gene expression prediction and cross-modal retrieval, thereby bridging pathology and molecular omics.

Note that some methods do not solely focus on pathology images but also encompass multi-modal medical imaging data such as Computed Tomography (CT), Magnetic Resonance Imaging (MRI), and X-ray from various organs (Zhang et al., 2023b; 2024; Zhao et al., 2024; Xia et al., 2024). However, since their goal is not to leverage other medical image modalities to enhance pathology image representation but rather to develop a universal medical image model, these studies exceed the scope of our survey.

## 4 Multi-Modal Datasets for CPath

Larger, more diverse, and higher-quality datasets for CPath have been proven to be the key to the success of FM4CPath (Vorontsov et al., 2023; Zimmermann et al., 2024), and MMFM4CPath is no exception. Curating pathology-specific public datasets has long been a challenge in this field, driving extensive research efforts. Many well-designed datasets have been developed to address various pathology-related questions, continuously advancing CPath. We summarize existing multi-modal datasets for CPath, highlighting high-

quality datasets or those that have demonstrated success in current models. Based on data types, we categorize them into three groups as shown in Table 3.

**Image-Text Pair Datasets for CPath.** This category includes tile-level tile-caption pairs and WSI-level WSI-report pairs. Training on these datasets within a self-supervised contrastive learning framework enables FM4CPath to learn richer image embeddings while gaining zero-shot and cross-modal capabilities. Due to the expensive expert annotation and the preference of many research institutions for in-house data, several datasets have been constructed by collecting pathology tile images and text data from online sources, books, and publicly available educational resources. For example, QUILT (Ikezogwo et al., 2024), OPENPATH (Huang et al., 2023), ARCH (Gamper & Rajpoot, 2021a) and MI-ZERO (Lu et al., 2023) leverage data from YouTube, Twitter, pathology textbooks, and educational resources, respectively. Due to the lack of a unified format, the collected data undergo standardized processing pipelines to ensure high quality. Image data is filtered for non-pathology images, followed by sub-figure segmentation. Text data is refined with LLMs, including sub-caption segmentation and token-based filtering. Finally, multimodal models align figures with captions. Additionally, QUILT (Ikezogwo et al., 2024) uses speech recognition to extract text from videos.

Other tile-caption pair datasets primarily expand existing datasets or utilize internal datasets to enhance scale and diversity (Sun et al., 2024e; Lu et al., 2024a; Sun et al., 2024d;b; Dai et al., 2024). Notably, ARCH (Gamper & Rajpoot, 2021a) is a multiple-instance captioning CPath dataset, where each image bag is associated with a single caption. Furthermore, datasets such as PATHGEN (Sun et al., 2024d), HISTGEN (Guo et al., 2024), and MASS-340K (Ding et al., 2024) generate WSI-report pairs by leveraging generative models or processing WSI descriptions using LLMs.

**Multi-Modal Instruction Datasets for CPath.** These datasets incorporate diverse instructions for tuning LLM-based vision-language FM4CPath, training them as AI assistants in the pathology domain. Since manually designing instructions is typically expensive, instruction construction often directly relies on LLMs to generate cost-effective instruction datasets. The most common instruction type is VQA, which typically includes closed-ended and open-ended question-and-answer (Q&A) sessions to develop the model's conversational abilities. Due to prompt flexibility, different datasets create various instructions based on their needs. For example, PATHINSTRUCT(Sun et al., 2024e) provides instruction-following samples that enable LLMs to call upon other pathology models for problem-solving. Lu *et al.*(Lu et al., 2024b) developed six instruction types to adapt the model to diverse pathology conversation scenarios. CLOVER INSTRUCTION(Chen et al., 2024a) generates instructions both through LLMs and by matching template questions with original text captions for cost-effectiveness. PATHMMU(Sun et al., 2024c) uses enhanced descriptions with images to prompt GPT-4V (GPT, 2023), generating professional multi-modal pathology Q&As with detailed explanations. Due to the scarcity of large-scale multi-modal pathology datasets for training WSI interpretation assistants, WSI-level instructions have emerged, typically creating VQAs from WSI reports and advanced LLM-generated prompts.

**Image-Other Modality Pair Dataset.** There is still a lack of extensive exploration of datasets involving vision and other modalities. Zhou *et al.* (Zhou et al., 2024a;b) constructed two different disease knowledge graphs and, guided by one of them, created well-structured semantic groups linked through hierarchical relations. The MBTG-47K dataset (Vaidya et al., 2025) includes paired data of DNA and RNA gene sequences with WSIs. Xu et al. (Xu et al., 2024) publicly released a dataset with WSI-report-RNA-sequence pairs containing three modalities. These are bold attempts at constructing datasets that integrate pathology images with other modalities.

# 5 Evaluation Tasks

## 5.1 Taxonomy of Evaluation Tasks in MMFM4CPath

Unlike uni-modal FM4CPath, data from other modalities not only enhance MMFM4CPath's understanding of pathology images but also enable MMFM4CPath to perform zero-shot learning and cross-modal tasks. When MMFM4CPath are combined with LLMs, they gain the ability to engage in dialogue and generation, allowing them to adapt to more diverse tasks. We have summarized the evaluation tasks used by MMFM4CPath, as shown in Figure 3, and categorized them into six main types from a machine learning

Table 3: Multi-Modal Datasets for CPath.

| Dataset† (Availability) | Data Type | Description | Staining‡ | Dataset Invariant | Data Source Public | Private | Method | LLM Assisted |
|---|---|---|---|---|---|---|---|---|
| **Image–Text Pair** | | | | | | | | |
| QUILT (Ikezogwo et al., 2024) ✓ | Tile-Caption Pair | 437,878 tiles paired with 802,144 captions extracted from 4,475 videos. | H, I, O | QUILT-1M: Combining QUILT with other pathology data sources to form 1M pairs. | YouTube | ✗ | QuiltNet (Ikezogwo et al., 2024) | ✓ |
| PATHCAP (Sun et al., 2024e) ✓ | Tile-Caption Pair | 207K pathology tile-caption pairs. | H, I, O | - | PubMed (Gu et al., 2021) | ✗ | PathCLIP (Sun et al., 2024e) | ✓ |
| OPENPATH (Sun et al., 2024e) ✓ | Tile-Caption Pair | 208,404 tile-caption pairs. | H, I, O | PATHLAION: 32,041 additional tile–caption pairs scraped from the Internet and the LAION dataset (Schuhmann et al., 2022) | WSI-Twitter, Replies, PATHLAION | ✗ | PLIP (Huang et al., 2023) | ✗ |
| CONCH* (Lu et al., 2024a) ✗ | Tile-Caption Pair | 1,170,647 tile–caption pairs. | H, I, O | - | PMC OA (Istrate et al., 2022) | ✓ | CONCH (Lu et al., 2024a) | ✓ |
| HISTGEN (Guo et al., 2024) ✓ | WSI-Report Pair | A WSI-report dataset with 7,753 pairs. | H | - | TCGA (Tomczak et al., 2015) | ✗ | - | ✓ |
| MASS-340K (Ding et al., 2024) ✗ | WSI | 335,645 WSIs across 20 organs. | H, I | Synthetic captioning for 423,122 ROIs and curation of 182,862 WSI-report pairs. | GTEx (Consortium et al., 2015) | ✓ | TITAN (Ding et al., 2024) | ✓ |
| CPATH-PATCH CAPTION (Sun et al., 2024b) ✗ | Tile-Caption Pair | 700,145 tile-caption pairs from diverse datasets. | H, I, O | - | PATHCAP, QUILT-1M, OPENPATH | ✗ | CPath-Omni (Sun et al., 2024b) | ✓ |
| PATHGEN (Sun et al., 2024d) ✓ | Tile-Caption Pair | 1.6 million high-quality tile-caption pairs from 7,300 WSIs. | H | PATHGEN$_{init}$: 700K tile-caption pairs from PATHCAP, OPENPATH, and QUILT-1M | TCGA (Tomczak et al., 2015) | ✗ | PathGen-CLIP (Sun et al., 2024d) | ✓ |
| MUNICH (Tran et al., 2025) ✗ | WSI-Report Pair | 15,129 paired WSIs and pathology reports from 6,705 patients. | H | - | - | ✓ | HistoGPT (Tran et al., 2025) | ✗ |
| PCAPTION-C (Tran et al., 2025) ✓ | Tile-Caption Pair | 1,409,058 tile-caption pairs. | H, I, O | PCAPTION-0.8M: removing non-human pathology data and PCAPTION-0.5M: further filter out pairs with <20 words. | PMC-OA (Istrate et al., 2022), QUILT-1M | ✗ | PA-LLaVA (Dai et al., 2024) | ✓ |
| ARCH (Gamper & Rajpoot, 2021a) ✓ | Bag-Caption Pair | 11,816 bags and 15,164 images, with each bag containing multiple tiles. | H, I | - | PubMed (Gu et al., 2021), pathology textbooks | ✗ | - | ✗ |
| MI-ZERO (Lu et al., 2023) ✓ | Tile-Caption Pair | Diverse dataset of 33,480 tile-caption pairs. | H, I, O | - | educational resources, ARCH | ✗ | - | ✗ |
| **Multi-Modal Instruction** | | | | | | | | |
| PATHINSTRUCT (Sun et al., 2024e) | Tile-Level Instruction | 180K pathology multi-modal instruction-following samples. | H, I, O | - | YouTube | ✗ | PathAsst (Lu et al., 2024b) | ✓ |
| CPATH-PATCH INSTRUCTION (Sun et al., 2024b) ✗ | Tile-Level Instruction | 351,871 tile-level samples, including tile-caption pairs, VQA pairs, labeled images for classification, and visual referring prompting pairs. | H | CPATH-VQA: created by generating VQA pairs using GPT-4o (Hurst et al., 2024), which combines classification labels with image data for datasets lacking captions. | CPATH-VQA, PATHGEN, CPATH-PATCHCAPTION, PATHINSTRUCT | ✓ | CPath-Omni (Sun et al., 2024b) | ✓ |
| CPATH-WSI INSTRUCTION (Sun et al., 2024b) ✗ | WSI-Level Instruction | 7,312 WSI-level samples, including captioning, VQA, and classification. | H | Further generate a WSI VQA dataset by prompting GPT-4 (Achiam et al., 2023). | HISTGEN | ✗ | CPath-Omni (Sun et al., 2024b) | ✓ |
| QUILT-INSTRUCT (Seyfioglu et al., 2024) ✓ | VQA Pair | 107,131 Q&A pairs. | H, I, O | QUILT-VQA: a Q&A dataset from Youtube videos, categorized into image-dependent and general-knowledge questions; QUILT-VQA-RED: QUILT-VQA with red circle marking the ROI in the pathology image. | YouTube | ✗ | Quilt-LLaVA (Seyfioglu et al., 2024) | ✓ |
| PathChat* (Lu et al., 2024b) ✗ | Tile-Level Instruction | 456,916 instructions with 999,202 question and answer turns. | H, I | PATHQABENCH: an expert-curated benchmark of 105 high-resolution pathology images, split into PATHQABENCH-PUBLIC and PATHQABENCH-PRIVATE subsets. | PMC-OA (Istrate et al., 2022), TCGA (Tomczak et al., 2015) | ✓ | PathChat (Lu et al., 2024b) | ✓ |
| CLOVER INSTRUCTION (Chen et al., 2024a) ✓ | Tile-Level Instruction | 45K question-and-answer instructions. | H | - | QUILT-VQA, PathVQA (He et al., 2020) | ✓ | CLOVER (Chen et al., 2024a) | ✓ |
| PATH-ENHANCEDS (Wu et al., 2024) ✓ | Tile-Level Instruction | 49K tile-level instructions, including captioning, VQA, classification and conversation. | H | - | OPENPATH, TCGA (Tomczak et al., 2015), PathVQA (He et al., 2020), etc. | ✗ | PathInsight (Wu et al., 2024) | ✓ |
| SLIDE-INSTRUCTION (Chen et al., 2024d) ✓ | WSI-Level Instruction | 44,181 WSI-caption pairs and 175,754 visual Q&A pairs. | H | SLIDEBENCH: 734 WSI captions along with a substantial number of closed-set VQA pairs to establish evaluation benchmark. | TCGA (Tomczak et al., 2015) | ✗ | SlideChat (Chen et al., 2024d) | ✓ |
| WSI-VQA (Chen et al., 2024b) ✓ | VQA Pair | 977 WSIs and 8,672 Q&A pairs. | H | - | TCGA-BRCA (Tomczak et al., 2015) | ✗ | W2T (Chen et al., 2024b) | ✓ |
| PA-LLaVA* (Dai et al., 2024) ✓ | VQA Pair | 35,543 question-answer pairs. | H | - | PathVQA (He et al., 2020) | ✗ | PA-LLaVA (Dai et al., 2024) | ✓ |
| WSI-Bench (Liang et al., 2024) ✗ | VQA Pair | 179,569 WSI-level VQA pairs, which span across 3 pathological capabilities with 11 tasks. | H | SLIDEBENCH: 734 WSI captions along with a substantial number of closed-set VQA pairs to establish evaluation benchmark. | TCGA (Tomczak et al., 2015) | ✗ | WSI-LLaVA (Liang et al., 2024) | ✓ |
| PATHMMU (Sun et al., 2024c) ✓ | VQA Pair | 33,428 Q&As along with 24,067 pathology images. | H, I, O | SLIDEBENCH: 734 WSI captions along with a substantial number of closed-set VQA pairs to establish evaluation benchmark. | PubMed (Gu et al., 2021), QUILT-1M, Atlas (Alber et al., 2025), OPENPATH | ✗ | - | ✓ |
| TCP-LLaVA* (Lyu et al., 2025) ✓ | VQA Pair | 175,797 Q&A pairs from 10 different cancer types. | H | - | TCGA (Tomczak et al., 2015), SLIDEBENCH, WSI-VQA | ✗ | TCP-LLaVA (Lyu et al., 2025) | ✗ |
| **Image-Other Modality** | | | | | | | | |
| KEEP* (Zhou et al., 2024a) ✓ | Pathology KG | KG contains 11,454 disease entities and 139,143 associated attributes. | - | - | DO (Schriml et al., 2012), UMLS (Bodenreider, 2004) | ✗ | KEEP (Zhou et al., 2024a) | ✓ |
| | Pathology Semantic Group | 143K pathology semantic groups linked through the disease KG | H, I, O | - | QUILT-1M, OPENPATH | ✗ | | |
| PATHKT (Zhou et al., 2024b) ✓ | Pathology KG | Pathology KG that consists of 50,470 informative attributes | - | - | OncoTree | ✗ | KEP (Zhou et al., 2024b) | ✗ |
| mSTAR* (Xu et al., 2024) ✓ | WSI-Report-RNA-Seq Pair | A dataset with 7,947 cases with image, text and RNA sequence modalities for pretraining. | H | - | TCGA (Tomczak et al., 2015) | ✗ | mSTAR (Xu et al., 2024) | ✓ |
| MBTG-47K (Vaidya et al., 2025) ✗ | WSI-RNA-Seq Pair WSI-DNA-Seq Pair | 26,615 WSI-RNA pairs, and 20,556 WSI-DNA pairs. | H | - | TCGA (Tomczak et al., 2015), GTEx (Consortium et al., 2015) | ✓ | THREADS (Vaidya et al., 2025) | ✗ |
| ST-BANK (Chen et al., 2025) ✗ | Tile–Transcriptomics Pair | 2,185,571 pathology-specific tile-transcriptomics pairs | H | - | 10x Visium (Mirzazadeh et al., 2023) | ✓ | OmiCLIP (Chen et al., 2025) | ✗ |

† Some methods introduced datasets without naming them, so we use the method name instead and marked with an asterisk (*).
‡ Staining type: **H**: H&E, **I**: IHC, **O**: Others.

perspective: classification, retrieval, generation, segmentation, prediction, and visual question answering (VQA). Furthermore, we classify them based on the type of pathology image input they target (tile or WSI). For MMFM4CPath, their pre-training dataset and model design are closely tied to their evaluation tasks.

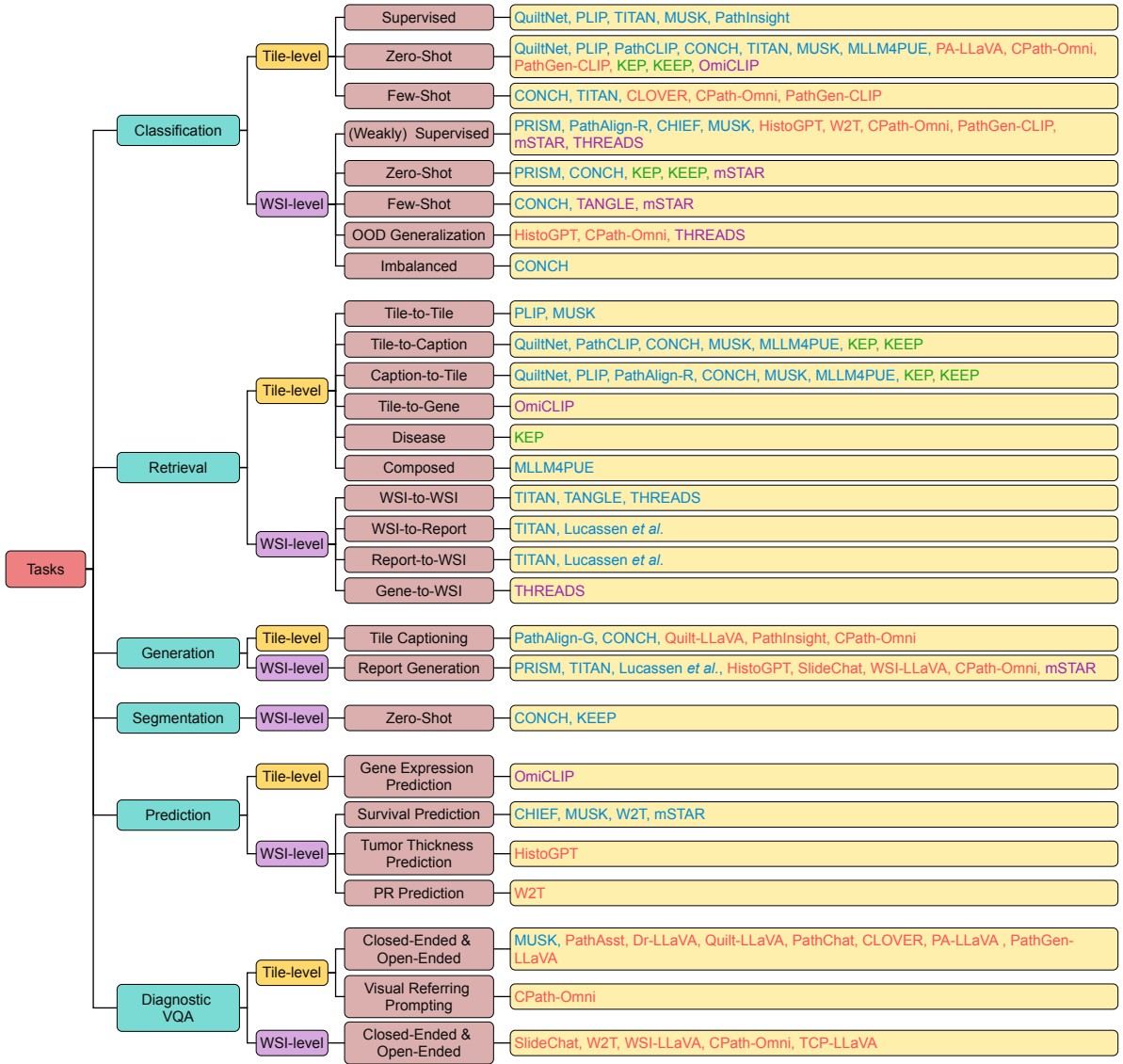

Figure 3: A comprehensive taxonomy of MMFM4CPath, categorized according to evaluation tasks. **Non-LLM-based vision-language**, **LLM-based vision-language**, **vision-knowledge graph**, and **vision-gene expression** models are highlighted in different colors, respectively.

For example, benefiting from multi-scale and more diverse training instructions, CPath-Omni (Sun et al., 2024b) has been evaluated across the widest range of tasks.

Classification is the most common evaluation task for MMFM4CPath, as many pathology-related tasks, such as cancer subtyping and biomarker prediction, are fundamentally classification problems. Most MMFM4CPath are evaluated on classification tasks across various settings. Models using tile-level inputs can perform WSI-level classification via multiple instance learning (MIL), treated as weakly supervised due to the lack of detailed region annotations. Multi-modal data enables zero-shot or few-shot classification with minimal reliance on costly annotations. Some methods also assess out-of-distribution (OOD) generalization to handle distribution shifts between training and test data (*e.g.*, data collected from different institutions). Additionally, CONCH (Lu et al., 2024a) evaluates classification on rare diseases with imbalanced data.

Table 4: Comparison of MMFM4CPath for zero-shot tile classification task on multiple pathology datasets. **Bold** is the best and underline is the second best.

| Method | WSSS4LUAD | | LC25000Lung | | LC25000Colon | | BACH | | CRC100K | | Osteo | | SICAPv2 | Pcam | SkinCancer | |
|---|---|---|---|---|---|---|---|---|---|---|---|---|---|---|---|---|
| | F1 | Accuracy | F1 | Accuracy | F1 | Accuracy | F1 | Accuracy | F1 | Accuracy | F1 | Accuracy | Accuracy | Accuracy | F1 | Accuracy |
| QuiltNet (Ikezogwo et al., 2024) | **0.827** | 0.705 | 0.775 | 0.800 | 0.910 | 0.943 | 0.386 | 0.438 | 0.553 | 0.495 | 0.585 | 0.538 | 0.373 | 0.587 | 0.409 | 0.464 |
| PLIP (Huang et al., 2023) | 0.261 | 0.731 | 0.622 | 0.879 | 0.902 | 0.902 | 0.380 | 0.343 | 0.602 | 0.528 | 0.322 | 0.529 | 0.425 | 0.518 | 0.435 | 0.425 |
| PathCLIP (Sun et al., 2024e) | - | 0.851 | - | 0.889 | **0.943** | 0.943 | - | 0.468 | - | 0.553 | - | 0.692 | 0.483 | 0.725 | - | 0.351 |
| CONCH (Lu et al., 2024a) | 0.798 | - | 0.527 | - | - | - | 0.606 | - | 0.590 | - | **0.785** | - | - | - | 0.413 | - |
| MLLM4PUE (Zhou et al., 2025) | 0.591 | - | 0.809 | - | 0.923 | - | 0.562 | - | 0.522 | - | 0.638 | - | - | - | 0.365 | - |
| CPath-Omni (Sun et al., 2024b) | - | **0.871** | - | **0.971** | - | **1.000** | - | **0.723** | - | **0.780** | - | **0.807** | 0.631 | **0.959** | - | **0.742** |
| PathGen-CLIP (Sun et al., 2024d) | - | 0.822 | - | 0.898 | - | 0.993 | - | 0.715 | - | **0.780** | - | 0.746 | **0.635** | 0.882 | - | 0.706 |
| KEEP (Zhou et al., 2024a) | 0.809 | - | **0.936** | - | - | - | **0.686** | - | **0.852** | - | 0.760 | - | - | - | **0.658** | - |

Table 5: Comparison of MMFM4CPath for tile-to-caption (image-to-text) and caption-to-tile (text-to-image) retrieval on multiple pathology datasets. **Bold** is the best and underline is the second best.

| Method | Tile-to-Caption (i2t) | | | | | | Caption-to-Tile (t2i) | | | | | |
|---|---|---|---|---|---|---|---|---|---|---|---|---|
| | Arch-PubMed | | | Arch-Book | | | Arch-PubMed | | | Arch-Book | | |
| | R@5 | R@10 | R@50 | R@5 | R@10 | R@50 | R@5 | R@10 | R@50 | R@5 | R@10 | R@50 |
| QuiltNet (Ikezogwo et al., 2024) | 0.069 | 0.111 | 0.273 | 0.116 | 0.168 | 0.384 | 0.056 | 0.092 | 0.237 | 0.100 | 0.152 | 0.389 |
| PLIP (Huang et al., 2023) | 0.037 | 0.067 | 0.185 | 0.096 | 0.152 | 0.393 | 0.037 | 0.067 | 0.181 | 0.112 | 0.164 | 0.419 |
| PathCLIP (Sun et al., 2024e) | 0.275 | 0.388 | 0.680 | 0.152 | 0.234 | 0.482 | 0.236 | 0.348 | 0.630 | 0.137 | 0.196 | 0.445 |
| MLLM4PUE (Zhou et al., 2025) | **0.372** | **0.495** | **0.782** | 0.192 | 0.283 | 0.603 | **0.297** | **0.399** | **0.688** | 0.185 | 0.277 | 0.555 |
| KEP (Zhou et al., 2024b) | - | 0.196 | 0.421 | - | 0.282 | 0.564 | - | 0.176 | 0.404 | - | 0.340 | 0.621 |
| KEEP (Zhou et al., 2024a) | 0.180 | 0.248 | 0.492 | **0.298** | **0.404** | **0.732** | 0.182 | 0.228 | 0.491 | **0.322** | **0.434** | **0.781** |

In addition to basic image-to-image retrieval, non-LLM-based MMFM4CPath are widely used for cross-modal retrieval tasks, such as text-to-image and image-to-text retrieval. KEP (Zhou et al., 2024b) performs one-to-many disease retrieval, retrieving captions or tiles with the same disease label using disease names. MLLM4PUE (Zhou et al., 2025) enables many-to-one composed retrieval by using pathology images and questions as queries. Moreover, due to its capability to understand gene expression data, THREADS (Vaidya et al., 2025) generates class prompts from gene expression profiles for WSI retrieval.

The integration of LLMs, whether by fine-tuning them as part of the model's architecture or by directly utilizing existing models, enables MMFM4CPath to generate captions/reports from tiles/WSIs. CONCH (Lu et al., 2024a) and KEP (Zhou et al., 2024b) evaluate the segmentation capabilities of these models. Some MMFM4CPath have also been tested for prediction tasks, using WSIs to generate continuous value predictions.

LLM-based MMFM4CPath models focus on evaluating their diagnostic VQA ability. Compared to traditional QA tasks, VQA incorporates pathology images into its questions, challenging the image understanding capabilities of V-LLMs. Typically, VQA tasks involve answers from a fixed set, usually in the form of closed-ended questions, such as multiple-choice (single or multiple answers) or true/false questions, as well as open-ended questions with no predefined answer options. These tasks can also be divided into multi- and single-turn dialogues. The initial LLM-based MMFM4CPath only performed tile-level VQA tasks (Sun et al., 2024e; Lu et al., 2024b), but recently, conversational abilities on WSI have gained increasing attention (Chen et al., 2024d; Liang et al., 2024). Additionally, CPath-Omni (Sun et al., 2024b) has been validated on the visual referring prompting task, where the regions of interest (ROIs) are highlighted, and both the question and answer are based on these regions. It is worth noting that, due to its flexible format, the VQA task offers high adaptability: tasks like classification and generation can be transformed into VQA tasks via prompt engineering (Wu et al., 2024; Sun et al., 2024b). Thus, LLM-based MMFM4CPath also encompass evaluation capabilities typical of non-LLM-based models. In addition to the quantitative analysis above, qualitative analysis is also frequently used to assess the performance of MMFM4CPath, especially their VQA and generation abilities. This is done by directly observing or through evaluation by professional pathologists to assess the quality of the generated text. For a more detailed discussion of the risks associated with automatic metrics and LLM-as-judge in generation tasks, as well as evidence from recent studies that employ pathologist-based evaluations, please see Appendix B.

## 5.2 Comparative Analysis of MMFM4CPath on Evaluation Tasks

Currently, research on MMFM4CPath lacks a unified benchmark that specifies a consistent protocol (e.g., evaluation datasets and metrics) for assessing different models. In this work, we have compared the evaluation results of all available methods covered in the survey as comprehensively as possible, in order to provide a useful reference for model selection. As noted in Section 5, although different studies may employ the same type of evaluation task, their choices of datasets and metrics are rarely standardized. To address this, we have collected two widely adopted and relatively consistent evaluation tasks and reported results across multiple datasets and metrics in Table 4 and Table 5.

Table 4 presents the F1-score and accuracy of eight MMFM4CPath models across nine datasets, namely WSSS4LUAD (Han et al., 2022), LC25000LUNG (Borkowski et al., 2019), LC25000COLON (Borkowski et al., 2019), BACH (Aresta et al., 2019), CRC100K (Kather et al., 2018), OSTEO (Arunachalam et al., 2019), SICAPv2 (Silva-Rodríguez et al., 2020), PCAM (Veeling et al., 2018), and SKINCANCER (Kriegsmann et al., 2022). The results indicate that even when evaluated on the same dataset and metric, different MMFM4CPath exhibit their own tendencies. Among them, CPath-Omni, PathGen-CLIP, and KEEP demonstrate superior and more robust performance across datasets. Table 5 shows the performance of six MMFM4CPath models on the ARCH-PUBMED (Gamper & Rajpoot, 2021b) and ARCH-BOOK (Gamper & Rajpoot, 2021b) datasets for tile-to-caption retrieval and caption-to-tile retrieval, measured by Top-K Recall (R@K, K = {5, 10, 50}). We observe that KEEP achieves the best results on ARCH-BOOK, PathCLIP shows stronger performance on ARCH-PUBMED, while MLLM4PUE delivers more robust and overall stronger retrieval ability across both datasets. Establishing a unified evaluation standard that specifies consistent tasks, datasets, and metrics for benchmarking different MMFM4CPath remains an urgent and important direction for future research, which we further discuss in Section 6.

## 6 Future Directions

**Developing MMFM4CPath Integrating H&E Images with Spatial Omics.** The integration of H&E-stained histopathology images with spatial omics data, such as spatial transcriptomics and proteomics, represents a promising frontier in computational pathology. By coupling morphological context with spatially resolved molecular signatures, future multi-modal foundation models could enable precise cellular localization of gene and protein expression, bridging the gap between tissue architecture and molecular mechanisms. Developing such models would require addressing challenges like data sparsity, spatial resolution mismatch, and alignment between modalities, but could significantly enhance our understanding of disease heterogeneity and microenvironmental interactions.

**Developing MMFM4CPath to Predict MxIF Markers from H&E Images.** A compelling direction involves using H&E images to predict marker expressions captured by multiplexed immunofluorescence (MxIF), enabling cost-effective and scalable estimation of protein-level biomarkers. This line of research leverages the morphological cues from H&E to infer high-dimensional proteomic data, potentially reducing the need for expensive MxIF experiments. Multi-modal foundation models trained with paired H&E-MxIF data could facilitate virtual staining or marker imputation, supporting downstream tasks such as subtyping, immune landscape assessment, and therapy response prediction in a non-invasive manner.

**Toward Multi-Modal Integration Beyond Pairs.** Most existing FM4CPath focus on pairwise modality alignment, such as vision–language or vision–gene expression. An important future direction is to extend this paradigm toward integration of three or more modalities (e.g., vision–language–gene expression or vision–knowledge graph–omics), which would enable more comprehensive modeling of pathological phenotypes by jointly capturing morphological, molecular, and semantic signals. While this higher-order integration currently faces significant challenges, including data sparsity, alignment complexity, and increased computational cost, it represents a natural evolution for the field. Progress in multi-modal pretraining frameworks, data harmonization, and scalable optimization techniques may gradually make such integration feasible.

**Advancing Trustworthy Multi-Modal CPathFMs.** An important yet underexplored direction is improving the *trustworthiness* of multi-modal foundation models in computational pathology. Beyond safeguarding patient privacy, especially when models incorporate clinical text or genomic data, future work must

also address fairness concerns (Shao et al., 2024; Lin et al., 2024), such as performance disparities (Li et al., 2024) arising from imbalanced datasets across demographics or disease subtypes. Incorporating privacy-preserving learning, rigorous de-identification, and fairness-aware evaluation will be essential for ensuring responsible and reliable deployment of multi-modal CPathFMs.

**Standardized Benchmarking for MMFM4CPath.** As the field matures, there is a pressing need to establish standardized metrics and unified benchmarks for evaluating MMFM4CPath. Current evaluations are fragmented across tasks, modalities, and datasets, limiting comparability and reproducibility. Future work should focus on developing comprehensive evaluation protocols that span classification, retrieval, generation, and VQA across tile- and WSI-level inputs. Such efforts would guide model development, ensure fair comparisons, and accelerate the translation of multi-modal models into clinical practice.

## 7 Conclusion

In this survey, we have systematically reviewed the recent advances in multi-modal foundation models for computational pathology, focusing on three major paradigms: vision-language, vision-knowledge graph, and vision-gene expression models. We categorized 34 state-of-the-art models, analyzed 30 multi-modal datasets, and summarized key downstream tasks and evaluation strategies. Our comprehensive comparison highlights the growing impact and promise of integrating diverse data modalities in computational pathology.

## Acknowledgment

Guihong Wan is supported by the National Cancer Institute of the National Institutes of Health (NIH) under Award Number K99CA286966. The work done by Yi He has been supported in part by the National Science Foundation (NSF) under Grant Numbers IIS-2505719, IIS-2441449, IOS-2446522, and the Commonwealth Cyber Initiative (CCI). The work done by Zhong Chen has been supported in part by an Illinois Innovation Network (IIN) sustaining Illinois seed funding grant.

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

## Appendix

## A   Survey Methodology

To ensure transparency and reproducibility of our survey, we briefly describe how the reviewed papers and datasets were identified and selected. We outline the literature databases searched, the main search terms used, the time period covered, and the inclusion/exclusion criteria applied. This methodological overview clarifies the scope of our survey and allows readers to assess its completeness and potential sources of bias.

- **Databases searched:** arXiv, medRxiv, bioRxiv, Google Scholar, ACM Digital Library, PubMed, DBLP, Europe PMC.

- **Search terms:** "computational pathology foundation model", "multi-modal computational pathology", "vision-language foundation model for computational pathology", "vision-omics foundation model for computational pathology", "large language model for computational pathology", "foundation model for computational pathology", etc.

- **Date range:** January 2022 – September 2025.

- **Inclusion/exclusion criteria:** This survey focuses on multi-modal foundation models (FMs) developed specifically for computational pathology (CPath), with an emphasis on models built upon hematoxylin and eosin (H&E) stained whole-slide images (WSIs) and tile-level representations. We review 34 state-of-the-art models that integrate pathology images with auxiliary modalities such as textual reports, knowledge graphs, and molecular profiles, categorizing them into vision–language, vision–knowledge graph, and vision–gene expression paradigms. In addition, we analyze 30 pathology-specific multi-modal datasets, grouped into image–text pairs, instruction datasets, and image–other modality pairs, and summarize the evaluation tasks and strategies most relevant to CPath foundation models. Several related directions are excluded from the scope of this survey. Specifically, methods that extend beyond pathology to broader biomedical imaging, including Computed Tomography (CT), Magnetic Resonance Imaging (MRI), and X-ray (Zhang et al., 2023b; 2024; Zhao et al., 2024; Xia et al., 2024), are not covered in detail, as their primary goal is to build universal medical imaging models rather than enhance pathology image representation. Similarly, we do not comprehensively review general-purpose multi-modal large language models (MLLMs) that incorporate pathology data only as a small subset of training, since their emphasis lies in broader generative AI capabilities rather than pathology-specific representation learning. By clearly defining these boundaries, we aim to provide a focused and coherent review of foundation models for computational pathology while acknowledging related but out-of-scope research directions.

## B   Risks of Automatic and LLM-based Evaluation in Generation Tasks

In generation tasks such as diagnostic VQA and pathology report generation, conventional automatic metrics (e.g., METEOR, BLEU, ROUGE) or even LLM-as-judge evaluations present several risks. Biases can arise because these metrics tend to reward surface-form fluency or frequent stylistic templates, leading to inflated scores for outputs that look plausible but misrepresent rare or clinically critical findings. Hallucinations, the fabrication of entities such as tumor grade, margin status, or molecular markers, may be overlooked by lexical overlap–based metrics and even tolerated by LLM judges if phrased fluently, despite being clinically unsafe. Dataset leakage and contamination pose further risks when near-duplicate reports or Q&A items exist across training and evaluation sets, or when LLM judges share training sources with the systems under evaluation, potentially inflating performance through circular validation. These limitations highlight the need for complementary expert review and contamination-aware evaluation protocols to ensure that generation tasks are assessed for clinical factuality rather than superficial textual similarity.

Beyond automatic and LLM-based metrics, several recent studies have incorporated pathologist-based evaluations for diagnostic VQA and report generation. For example, PathChat was evaluated on 260 open-ended diagnostic VQA questions by a panel of seven board-certified pathologists, who graded the correctness and

ranked the quality of model responses, revealing substantial gaps between human judgment and automatic metrics. Similarly, HistoGPT, a model for whole-slide report generation, was assessed by certified dermatopathologists, who judged whether generated reports captured the essential clinical findings and flagged factual inaccuracies. Both studies underscore that human experts prioritize clinical factuality and actionability over surface overlap, frequently identifying omission errors such as missing tumor stage, commission errors such as hallucinated findings, or semantic inversions such as "invasive" vs. "in situ." While pathologist review is costly and less scalable, it remains the most reliable safeguard against subtle yet clinically critical errors. Taken together, these observations highlight the necessity of integrating structured expert review into future benchmarks, where automatic metrics and human-in-the-loop evaluation should be reported jointly to provide a more faithful assessment of generative models in computational pathology.

