# OpenReview forum: "Multi-Modal Foundation Models for Computational Pathology: A Survey"
_TMLR — Accepted by TMLR_

### Review · Reviewer_9HxA · 2025-09-14

**Summary Of Contributions:**

The authors list, describe and organize multi-modal foundation models and datasets for computational pathology, which is concerned with using ML and other computational techniques to understand, classify and organize diseased tissues, often by analyzing images of stained tissues. The paper organizes these foundation models into 3 broad types, depending on which pair of modalities are combined: vision-language, vision-knowledge graph and vision-genomic. There is also a distinction made between LLM and non-LLM language modeling Architectures and other modelling detailed of these foundation models are described. The various tasks the models can be used for (classification, prediction,  retrieval, generative applications, etc) , Datasets are organized into image-text, instruction, and image-other modality categories. Future directions are suggests, such as combining these methods with spatial omics so as to understand the locality of the phenotypic consequences of gene expression in particular areas of tissue.

Although I'm neither an expert in computational pathology nor in foundation models, my impression is that this survey is quite thorough and comprehensive and the subcategorization of methods should be useful to various researchers.

The weaknesses I noticed were fairly minor. The paper seems to me to be a little light on analysis of trends of what seems to work well in this area (although perhaps that's because of the lack of consistent evaluation standards in the field, noted by the authors at the end of the paper). The paper leans a little bit too much in the direction of being a laundry list of models and datasets for my taste, at the expense of analysis of what's most important and what seems to work. However, I suppose a survey paper is not the same as a meta-analysis, so I think the paper generally achieves the goals of a survey paper.

The future directions mentions leaves out what (to me) seems to be an obvious future direction: combining 3 or more modalities (e.g. vision-language-gene expression or vision-knowledge graph-gene expression).

There are also a few typos which I will list below.

**Audience:**

Yes

**Audience Explanation:**

Computational pathology is a well-established and important field. Foundation models are a fast-moving and highly active field. As such, it is worthwhile to to collect current research into a survey paper, as the authors have done. This paper will help researchers entering the field to get caught up quickly.

**Broader Impact Concerns:**

I don't have serious concerns here.  I suppose that there could be some risk that poorly designed LLMs could reveal confidential details of patients inappropriately ? Maybe some discussion of that would be helpful. I think the paper is OK as-is, though.

**Claims And Evidence:**

Yes

**Claims Explanation:**

Claims are not really at the center of what makes a survey paper valuable, but the claim that the field of multi-modal foundation model computational pathology is organized into vision-language, vision-knowledge-graph and vision-gene expression seems well supported, with many specific examples of each subtype provided.

**Requested Changes:**

Some discussion of the possibility of 3-or-more-modalities as a future direction would be nice. Maybe that idea is unrealistic in the near future, but it's such an obvious idea that I think the paper would benefit from some discussion of it, regardless.

A couple of typos:

In Table 1, Instrution should be Instruction

At the bottom of page 9, I'm pretty sure 'VAQ' should be 'VQA'.

The first sentence of Section 2.2 on page 4 should have a comma rather than a period following "(SSCL)".

---

> ### Author Response · Authors · 2025-09-30
> **Response to Reviewer 9HxA**
>
> We thank the reviewer for the thoughtful and constructive feedback. We are pleased that the reviewer found our survey to be thorough, comprehensive, and useful for researchers in computational pathology. Below we address the points raised:
>
> > 1. Little quantitative comparison between difference methods
>
> Thank you for raising this point! Due to the diversity of tasks, evaluation datasets, and heterogeneous evaluation protocols, establishing standardized benchmarks in this field remains an urgent research direction, which we have also discussed in Section 6. We have compared the results of all the methods covered in this survey across different evaluation tasks. We note that although different papers may adopt the same type of evaluation task, the choice of datasets and metrics is rarely standardized. In the revised manuscript, we have added Section 5.2, Table 4, and Table 5 to provide a detailed discussion and analysis of this issue. Specifically, we collected two widely adopted and relatively consistent evaluation tasks, tile-level zero-shot classification and tile-level retrieval, and reported the results of multiple MMFM4CPath  across different datasets and metrics in Table 4 and Table 5, respectively, as a reference for model selection. Please see Section 5.2 for details.
>
> > 2. Future directions: beyond two modalities
>
> We thank the reviewer for pointing out the omission of three-or-more modality integration. While such integration faces practical barriers, we agree that it is a natural and important future direction. We have now added a new paragraph in the *Future Direction: Toward Multi-Modal Integration Beyond Pairs* section explicitly discussing this possibility. The added paragraph reads as follows:
>
> _Most existing FM4CPath focus on pairwise modality alignment, such as vision–language or vision–gene expression. An important future direction is to extend this paradigm toward integration of three or more modalities (e.g., vision–language–gene expression or vision–knowledge graph–omics), which would enable more comprehensive modeling of pathological phenotypes by jointly capturing morphological, molecular, and semantic signals. While this higher-order integration currently faces significant challenges, including data sparsity, alignment complexity, and increased computational cost, it represents a natural evolution for the field. Progress in multi-modal pretraining frameworks, data harmonization, and scalable optimization techniques may gradually make such integration feasible._
>
> > 3. Typos and minor corrections
>
> Thank you for your careful reading and corrections. We have corrected them in our paper as follows:
> + Table 1: “Instrution” → “Instruction”.
> + Bottom of Page 9: “VAQ” → “VQA”.
> + Page 4, Section 2.2: replaced the period with a comma after “(SSCL)”.
>
> > 4. Broader impact considerations about the patient privacy
>
> This is an excellent consideration. Multi-modal foundation models, particularly those involving large language models, must carefully safeguard patient privacy. However, to date no prior work has explicitly studied this issue. We have now added a new paragraph in the *Future Direction: Safeguarding Patient Privacy in Multi-Modal Foundation Models* section explicitly discussing this point, emphasizing that once multi-modal models involve patient-level textual records or genomic data, additional attention must be paid to de-identification and regulatory compliance. The added paragraph reads as follows:
>
> _Another important but underexplored direction is the safeguarding of patient privacy. When multi-modal models incorporate patient-level textual records or genomic data, they inevitably involve sensitive and identifiable information. Yet, existing research has rarely examined this challenge in depth. Future work should therefore adopt de-identification strategies, privacy-preserving learning frameworks, and compliance with healthcare regulations to ensure responsible use in computational pathology._

---

### Review · Reviewer_Qd8m · 2025-09-15

**Summary Of Contributions:**

**Summary:**

This survey reviews multimodal foundation models for computational pathology, focusing on H&E whole-slide images and tile-level inputs. It does four things. First, it groups thirty-two recent models into three families: vision and language (both non-LLM and LLM based), vision and knowledge graph, and vision and gene expression, and compares their architectures, pretraining goals, and adaptation strategies. Second, it compiles twenty-eight datasets, including image-text pairs, instruction datasets, and image-other-modality pairs. Third, it proposes a task taxonomy covering classification, retrieval, generation, segmentation, prediction, and visual question answering at both tile and WSI levels. Fourth, it highlights future directions such as integrating spatial omics, predicting multiplexed immunofluorescence markers from H&E, and creating standardized benchmarks. The survey also provides a visual roadmap of models (Fig. 1), a comparison to prior surveys highlighting broader coverage (Table 1), a large model pretraining/architecture table (Table 2), a dataset catalogue (Table 3), and a task taxonomy diagram (Fig. 3).

**Strengths:**
- S1: The work covers image-and-text models, and it also includes models that link images to medical knowledge graphs and to gene-expression data. It reviews 32 models and 28 datasets, which is quite exhaustive.
- S2: This survey clearly explains how models are trained (for example, CLIP, CoCa, BLIP-2, iBOT), how instruction tuning is done, and how they are evaluated (zero-shot or few-shot, WSI or tile), which makes it easier to compare papers that use different setups.
- S3: The paper groups evaluations into six task types (classification, retrieval, generation, segmentation, prediction, and VQA) and shows whether results are at the tile level or the whole-slide image (WSI) level, with Figure 3 and the nearby text guiding new readers.
- S4: The paper lists useful resources, including image and text pairs, instruction datasets, and collections that link images with other data such as pathology knowledge graphs and WSI with omics. This provides readers an easy starting point to find training data and benchmarks.

**Weaknesses:**
- W1: The survey is well organized, but it has little quantitative comparison across common tasks and datasets. Since it calls for standard benchmarks, add a small, curated table with harmonized metrics on datasets such as SlideBench, WSI VQA, or TCGA, even if the comparisons are imperfect.
- W2: VQA and report generation are scored mainly by automatic metrics or an LLM-as-judge, and although pathologist reviews are mentioned, the paper lacks a discussion on biases, hallucinations, and dataset leakage.
- W3: Please include the works _[1,2,3]_ and give the manuscript a careful proofread to remove typos and improve consistency.
- W4: It’s not explained how the papers were found and chosen, so readers can’t judge completeness or potential bias.
---
_[1] Chen, Weiqing, Pengzhi Zhang, Tu N. Tran et al. "A visual–omics foundation model to bridge histopathology with spatial transcriptomics." Nature Methods (2025): 1-15._

_[2] Lyu, Weimin, Qingqiao Hu, Kehan Qi et al. "Efficient Whole Slide Pathology VQA via Token Compression." arXiv preprint arXiv:2507.14497 (2025)._

_[3] Tran, Manuel, Paul Schmidle, Ruifeng Ray Guo et al. "Generating dermatopathology reports from gigapixel whole slide images with HistoGPT." Nature Communications 16, no. 1 (2025): 4886._

**Audience:**

Yes

**Audience Explanation:**

This paper offers a clear, well-organized survey that can orient both newcomers and practitioners. It also highlights concrete future directions and benchmarking needs, making it a handy reference for researchers building models or datasets in this area.

**Broader Impact Concerns:**

Not very applicable here.

**Claims And Evidence:**

Yes

**Claims Explanation:**

The survey paper’s claims are supported by clear tables and figures, including a catalog of 32 models and 28 datasets and a well-structured task taxonomy.

**Requested Changes:**

1. Add a small table with consistent metrics on common datasets such as SlideBench, WSI VQA, and TCGA to complement the narrative taxonomy.
2. Provide a brief Methods section listing databases, search terms, date range, and inclusion/exclusion rules, so readers can assess completeness and bias.
3. Analyze risks with automatic metrics and LLM-as-judge; the paper currently notes qualitative, pathologist-based checks without deeper analysis.
4. Specify metric choices and default data splits to improve cross-paper comparability alongside the task taxonomy.
5. Please include the works [1,2,3] in your figures and tables.
6. Correct typos: "visio-language" (in CoCa-based Vision-Language FM4CPath), "VAQs" (in Multi-Modal Instruction Datasets for CPath), "based on the these regions" (in Evaluation Tasks), "Instrution" (Table 1), "ECVA" (Table 2 footnote), "BILP-2" (Table 2, CLOVER row), "Qulit" (multiple places), "TGCA" (Table 3).

---

> ### Author Response · Authors · 2025-09-30
> **Response to Reviewer Qd8m**
>
> We sincerely thank the reviewer for the thorough and constructive feedback. We appreciate the recognition of our survey’s breadth, clarity, and usefulness, and we address each of the requested changes below.
>
> > **[W1, RC1 & RC4]**: Add a quantitative comparison across common tasks and datasets
>
> Given the diversity of tasks, evaluation datasets, and different evaluation protocols, establishing a standardized benchmark in this field is indeed an important direction for future research, which we have also discussed in Section 6. We have compared, as comprehensively as possible, the results of all methods covered in this paper across different evaluation tasks. We noticed that, although some papers share the same evaluation tasks, the datasets and metrics they adopt are often inconsistent, making direct comparison difficult.
>
> In the revised manuscript, we have added Section 5.2, Table 4, and Table 5 to provide a detailed discussion and analysis of this issue. Specifically, we collected two widely adopted evaluation tasks with relatively standardized settings, tile-level zero-shot classification and tile-level retrieval, and reported the results of multiple MMFM4CPath models under different datasets and metrics in Table 4 and Table 5, respectively. These results are intended to serve as a reference for model selection. For more details, please refer to Section 5.2.
>
> > **[W4, RC2]**: Provide a brief Methods section on how papers were found and chosen
>
> We have included the following information in the *Survey Methodology* section of the Appendix, covering our literature search strategy, including databases, search terms, time range, and inclusion/exclusion criteria, to improve transparency and allow readers to assess completeness and potential bias. Due to the word limit of this response, please refer to the revised manuscript for details.
>
> > **[W2, RC3]**: Analyze risks with automatic metrics and LLM-as-judge
>
> We thank the reviewer for highlighting the limitations of relying solely on automatic metrics or LLM-as-judge for evaluating generation tasks. In the revised manuscript, we have added a dedicated subsection in the Appendix B titled *“Risks of Automatic and LLM-based Evaluation in Generation Tasks.”* This section explicitly analyzes the risks of bias, hallucination, and dataset leakage, and discusses why these issues can cause automatic metrics and LLM-based scoring to misrepresent clinical correctness. Furthermore, we now include concrete examples of pathologist-based evaluations from recent studies (PathChat for open-ended VQA and HistoGPT for whole-slide report generation), illustrating how domain experts identify omission, commission, and semantic errors that are often overlooked by automatic scoring. These additions provide the requested deeper analysis and emphasize the necessity of integrating expert-in-the-loop evaluation into future benchmarks.
>
> > **[W3, RC5]**: Include missing works [1,2,3]
>
> We sincerely thank you for helping us identify the missing works. We have incorporated works [1,2] into our figures, tables, and main text. As for [3], it had already been included in our manuscript, and we have now updated the corresponding content based on its latest published version.
>
> > **[RC6]**: Correct typos and improve consistency
>
> Thank you for your careful reading and corrections. We have carefully proofread the manuscript and corrected the following issues:
> + “visio-language” → “vision-language” in subsection “CoCa-based Vision-Language FM4CPath”
> + “VAQs” → “VQAs” in subsection “Multi-Modal Instruction Datasets for CPath”
> + “based on the these regions” → “based on these regions” in Section 5
> + “Instrution” → “Instruction” in Table 1
> + “ECVA” → “ECCV” in Figure 1
> + “BILP-2” → “BLIP-2” in Table 2
> + “Qulit” → “Quilt” in multiple places
> + “TGCA” → “TCGA” in Table 3
>
> ---
>
> *[1] Chen et al., Nature Methods (2025): Visual–omics foundation model bridging histopathology and spatial transcriptomics.*
>
> *[2] Lyu et al., arXiv:2507.14497 (2025): Efficient whole-slide VQA via token compression.*
>
> *[3] Tran et al., Nature Communications (2025): HistoGPT for dermatopathology report generation.*

---

### Review · Reviewer_8caN · 2025-09-17

**Summary Of Contributions:**

The authors provide a review of large-scale (ie. "foundation model) deep-learning methods for computational pathology. In particular, this work provides a taxonomy of models and training methods for these tasks.

**Additional Comments:**

Due to my unfamiliarity with the field of computational pathology, I am unable to provide a strong recommendation for this work. However, it does appear to provide a fairly comprehensive coverage within this domain of foundation/ssl-model techniques that are familiar to me.

**Audience:**

Yes

**Audience Explanation:**

Yes, the content of this work is clearly relevant for certain groups of ML practitioners, as well as researchers interested in foundation models, since they are often benchmarked across many modalities.

**Claims And Evidence:**

Yes

**Claims Explanation:**

I am not familiar with the field of computational pathology, but at least to me it seems that it provides an extensive list and description of other studies and datasets, tabulated with annotations about methodology and public availability. The work also provides an overview of the different methods, often focusing on LLMs or models trained with cross-modal contrastive learning (ie. visual-language models, like CLIP).

**Requested Changes:**

The article does not discuss in sufficient detail the performance of these foundation models in practical tasks. Given that the purpose of foundation models is to be pretrained on a large body of data to improve performance on the actual desired task, I am wondering (as, again, I am unfamiliar with the field) if it is possible to include what the performance of the referenced models are on various downstream tasks, so that readers can get an overview of how well the different approaches do. For example, is there a standard benchmark task (or set of tasks) that papers commonly report performance on? If so, it may be useful including their reported performance. If not, it may still be valuable for a survey to inform readers of the scope and/or limitations of the other works referenced here, for a better broad picture of the state of the field (since Fig. 3 already includes information about which models were used for what tasks, it seems reasonable to suggest that the performance on each task be listed).

Including information about performance might be important because self-supervised learning and foundation models are often employed to improve performance over direct training on the desired task, which means that readers would obviously be interested to see this information. Because of this, it may be convenient to review some works that use approaches without foundation models as a "baseline" for progress in the field.

---

> ### Author Response · Authors · 2025-09-30
> **Response to Reviewer 8caN**
>
> We appreciate the reviewer’s feedback and the recognition of the value of our survey. We agree with the reviewer’s main concern that the manuscript currently lacks a consolidated discussion of performance on downstream tasks, which is indeed important for readers to appreciate the practical value of different foundation models.
>
> Due to the diversity of tasks, evaluation datasets, and heterogeneous evaluation protocols, establishing standardized benchmarks in this field remains an important and urgent research direction, which we have also discussed in Section 6. To provide readers with a more consistent view, we have compared the results of all methods covered in this survey across different evaluation tasks. We note that although different studies may adopt the same type of evaluation task, their choices of datasets and metrics are rarely standardized. In the revised manuscript, we have added Section 5.2, along with Table 4 and Table 5, to provide a detailed discussion and analysis of this issue. Due to the word limit for the response, we only provide Table 4 and Table 5 here for the reviewers’ reference. For a detailed analysis, please refer to Section 5.2 of the revised version of the paper.
>
>
> ### Table 4: Comparison of MMFM4CPath for zero-shot tile classification task on multiple pathology datasets
> **Bold** is the best and _underline_ is the second best.
>
> | Method | WSSS4LUAD F1 | WSSS4LUAD Acc | LC25000Lung F1 | LC25000Lung Acc | LC25000Colon F1 | LC25000Colon Acc | BACH F1 | BACH Acc | CRC100K F1 | CRC100K Acc | Osteo F1 | Osteo Acc | SICAPv2 Acc | Pcam Acc | SkinCancer F1 | SkinCancer Acc |
> |--------|--------------|---------------|----------------|-----------------|-----------------|------------------|---------|----------|------------|-------------|----------|-----------|--------------|----------|----------------|----------------|
> | QuiltNet | **0.827** | 0.705 | 0.775 | 0.800 | 0.910 | 0.943 | 0.386 | 0.438 | 0.553 | 0.495 | 0.585 | 0.538 | 0.373 | 0.587 | 0.409 | 0.464 |
> | PLIP | 0.261 | 0.731 | 0.622 | 0.879 | 0.902 | 0.902 | 0.380 | 0.343 | _0.602_ | 0.528 | 0.322 | 0.529 | 0.425 | 0.518 | _0.435_ | 0.425 |
> | PathCLIP | - | _0.851_ | - | 0.889 | **0.943** | 0.943 | - | 0.468 | - | 0.553 | - | 0.692 | 0.483 | 0.725 | - | 0.351 |
> | CONCH | 0.798 | - | 0.527 | - | - | - | _0.606_ | - | 0.590 | - | **0.785** | - | - | - | 0.413 | - |
> | MLLM4PUE | 0.591 | - | _0.809_ | - | _0.923_ | - | 0.562 | - | 0.522 | - | 0.638 | - | - | - | 0.365 | - |
> | CPath-Omni | - | **0.871** | - | **0.971** | - | **1.000** | - | **0.723** | - | **0.780** | - | **0.807** | _0.631_ | **0.959** | - | **0.742** |
> | PathGen-CLIP | - | 0.822 | - | _0.898_ | - | _0.993_ | - | _0.715_ | - | **0.780** | - | _0.746_ | **0.635** | _0.882_ | - | _0.706_ |
> | KEEP | _0.809_ | - | **0.936** | - | - | - | **0.686** | - | **0.852** | - | _0.760_ | - | - | - | **0.658** | - |
>
>
> ### Table 5: Comparison of MMFM4CPath for tile-to-caption (image-to-text) and caption-to-tile (text-to-image) retrieval on multiple pathology datasets
> **Bold** is the best and _underline_ is the second best.
>
> | Method | Arch-PubMed R@5 | Arch-PubMed R@10 | Arch-PubMed R@50 | Arch-Book R@5 | Arch-Book R@10 | Arch-Book R@50 | Arch-PubMed R@5 | Arch-PubMed R@10 | Arch-PubMed R@50 | Arch-Book R@5 | Arch-Book R@10 | Arch-Book R@50 |
> |--------|-----------------|------------------|------------------|---------------|----------------|----------------|-----------------|------------------|------------------|---------------|----------------|----------------|
> | QuiltNet | 0.069 | 0.111 | 0.273 | 0.116 | 0.168 | 0.384 | 0.056 | 0.092 | 0.237 | 0.100 | 0.152 | 0.389 |
> | PLIP | 0.037 | 0.067 | 0.185 | 0.096 | 0.152 | 0.393 | 0.037 | 0.067 | 0.181 | 0.112 | 0.164 | 0.419 |
> | PathCLIP | _0.275_ | _0.388_ | _0.680_ | 0.152 | 0.234 | 0.482 | _0.236_ | _0.348_ | _0.630_ | 0.137 | 0.196 | 0.445 |
> | MLLM4PUE | **0.372** | **0.495** | **0.782** | _0.192_ | _0.283_ | _0.603_ | **0.297** | **0.399** | **0.688** | _0.185_ | 0.277 | 0.555 |
> | KEP | - | 0.196 | 0.421 | - | 0.282 | 0.564 | - | 0.176 | 0.404 | - | _0.340_ | _0.621_ |
> | KEEP | 0.180 | 0.248 | 0.492 | **0.298** | **0.404** | **0.732** | 0.182 | 0.228 | 0.491 | **0.322** | **0.434** | **0.781** |
>
> We believe these additions will address the reviewer’s concern and provide readers with a clearer overview of how foundation models perform in practice.

---

### Author Response · Authors · 2025-10-01
**Author Response and Revised Version Summary**

We would like to sincerely thank you again for the time and effort you have devoted to reviewing our manuscript. We have now released our point-by-point responses along with a revised version of the paper.

The revised version incorporates several modifications, including:
+ According to the suggestions of the three reviewers, we have added Section 5.2 to discuss *Comparative Analysis of MMFM4CPath on Evaluation Tasks*, and we have included quantitative comparisons across metrics and datasets for two relatively standardized tasks (adding Table 4 and Table 5), in order to provide readers with a clearer overview of how foundation models perform in practice.
+ According to Reviewer Qd8m’s suggestion, we have incorporated two new works into the manuscript: the LLM-based vision–language model TCP-LLaVA [1] and the vision–GE model OmiCLIP [2]. To this end, we have made the following revisions:
  + Included the two new works in the roadmap of Figure 1.
  + Updated the number of models and datasets surveyed in Table 1, and revised the corresponding counts in other parts of the manuscript (the abstract, the contributions in Section 1, and the conclusion) to clarify the contribution of this survey.
  + Added the two methods to Table 2.
  + Discussed the newly added methods in Section 3.2 and Section 3.3.
  + Added the two corresponding datasets to Table 3.
  + Adjusted Figure 3 to reflect the evaluation tasks introduced by the new works.
+ According to Reviewer Qd8m’s suggestion, we have updated the corresponding content in the manuscript based on the latest release version of HistoGPT [3].
+ According to Reviewer 9HxA’s suggestion, we have further expanded the discussion in Section 6 on the two future directions: *Toward Multi-Modal Integration Beyond Pairs* and *Safeguarding Patient Privacy in MMFM4CPath*.
+ According to Reviewer Qd8m’s suggestion, we have added a *Survey Methodology* section in the appendix to present our literature search strategy, including databases, search terms, time range, and inclusion/exclusion criteria, in order to improve transparency and allow readers to assess completeness and potential bias. We have also added a corresponding reference in Section 1, paragraph 4 of the main text.
+ According to Reviewer Qd8m’s suggestion, we have added a section in the appendix entitled *“Risks of Automatic and LLM-based Evaluation in Generation Tasks,”* where we discuss the risks of bias, hallucination, and dataset leakage that may arise from automatic metrics and LLM-based scoring, and analyze these issues with illustrative examples from pathologist-based evaluations. We have also added a corresponding reference in Section 5.1, paragraph 5 of the main text.
+ According to the feedback from Reviewer 9HxA and Reviewer Qd8m, we have corrected a series of typos in the manuscript:
  + “visio-language” → “vision-language” in subsection “CoCa-based Vision-Language FM4CPath”
  + “VAQs” → “VQAs” in subsection “Multi-Modal Instruction Datasets for CPath”
  + “based on the these regions” → “based on these regions” in Section 5
  + “Instrution” → “Instruction” in Table 1
  + “ECVA” → “ECCV” in Figure 1
  + “BILP-2” → “BLIP-2” in Table 2
  + “Qulit” → “Quilt” in multiple places
  + “TGCA” → “TCGA” in Table 3
  + Replace the period with a comma after “(SSCL)” in Section 2.2

We would like to once again thank the reviewers for their careful reading, constructive feedback, and recognition of the value of our survey. We hope that these revisions adequately address your comments and strengthen the clarity, completeness, and impact of the survey. We remain open to further suggestions, and we would be very glad to continue the discussion if you have any additional feedback.

---

*[1] Chen et al., Nature Methods (2025): Visual–omics foundation model bridging histopathology and spatial transcriptomics.*

*[2] Lyu et al., arXiv:2507.14497 (2025): Efficient whole-slide VQA via token compression.*

*[3] Tran et al., Nature Communications (2025): HistoGPT for dermatopathology report generation.*

---

### Decision · Action_Editor_34Ui · 2025-11-07

**Recommendation:** Accept as is

**Audience:**

Yes

**Audience Explanation:**

The field of computational pathology is rapidly growing, and this survey will be valuable to both practitioners and new researchers. It provides a well-structured introduction to the core models and datasets, as well as task taxonomies and future research directions. The paper gives an up-to-date overview and is likely to serve as a useful reference in this area. This work will be relevant to a wider section of researchers who are interested in developing foundational models in diverse domains.

**Claims And Evidence:**

Yes

**Claims Explanation:**

This paper presents an up-to-date survey of multi-modal foundation models in computational pathology. The survey is comprehensive, well-organized and well-written, covering a broad selection of multi-modal foundation models for computational pathology. The authors have addressed all reviewer feedback by including benchmark comparisons, a methods summary, and evaluation risks. They have also improved the manuscript’s clarity and completeness by correcting typos and providing additional context for key issues such as privacy and future trends beyond pairwise modality integration. Overall, the technical presentation is clear, the content is thorough, and the survey will be a useful resource for the TMLR audience. I recommend acceptance.